# SITCOM: Step-wise Triple-Consistent Diffusion Sampling For Inverse Problems

**Ismail R. Alkhouri** [* 1 2]  **Shijun Liang** [* 3]

**Cheng-Han Huang** [2]  **Jimmy Dai** [1]  **Qing Qu** [1]  **Saiprasad Ravishankar** [2 3]  **Rongrong Wang** [2 4]

## Abstract

Diffusion models (DMs) are a class of generative models that allow sampling from a distribution learned over a training set. When applied to solving inverse problems, the reverse sampling steps are modified to approximately sample from a measurement-conditioned distribution. However, these modifications may be unsuitable for certain settings (e.g., presence of measurement noise) and non-linear tasks, as they often struggle to correct errors from earlier steps and generally require a large number of optimization and/or sampling steps. To address these challenges, we state three conditions for achieving measurement-consistent diffusion trajectories. Building on these conditions, we propose a new optimization-based sampling method that not only enforces standard data manifold measurement consistency and forward diffusion consistency, as seen in previous studies, but also incorporates our proposed step-wise and network-regularized backward diffusion consistency that maintains a diffusion trajectory by optimizing over the input of the pre-trained model at every sampling step. By enforcing these conditions (implicitly or explicitly), our sampler requires significantly fewer reverse steps. Therefore, we refer to our method as **S**tep-wise **T**riple-**Con**sistent Sa**m**pling (**SITCOM**). Compared to SOTA baselines, our experiments across several linear and non-linear tasks (with natural and medical images) demonstrate that SITCOM achieves competitive or superior results in terms of standard similarity metrics and run-time.

---

[*]Equal contribution [1]Electrical Engineering & Computer Science at University of Michigan–Ann Arbor [2]Computational Mathematics, Science, & Engineering at Michigan State University (MSU) [3]Biomedical Engineering at MSU [4]Mathematical Sciences at MSU. Correspondence to: Ismail R. Alkhouri <ismailal@umich.edu; alkhour3@msu.edu>.

*Proceedings of the 42nd International Conference on Machine Learning*, Vancouver, Canada. PMLR 267, 2025. Copyright 2025 by the author(s).

## 1. Introduction

Inverse problems (IPs) arise in a wide range of science and engineering applications, including computer vision (Li et al., 2024), signal processing (Byrne, 2003), medical imaging (Alkhouri et al., 2024b), remote sensing (Levis et al., 2022), and geophysics (BniLam & Al-Khoury, 2020). In these applications, the primary goal is to recover an unknown image or signal $\mathbf{x} \in \mathbb{R}^n$ from its measurements or a degraded image $\mathbf{y} \in \mathbb{R}^m$, which are often corrupted by noise. Mathematically, the unknown signal and the measurements are related as

$$\mathbf{y} = \mathcal{A}(\mathbf{x}) + \mathbf{n} \,,$$

where $\mathcal{A}(\cdot) : \mathbb{R}^n \to \mathbb{R}^m$ (with $m \leq n$) represents the linear or non-linear forward operator that models the measurement process, and $\mathbf{n} \in \mathbb{R}^m$ denotes the noise in the measurement domain, e.g., assumed sampled from a Gaussian distribution $\mathcal{N}(\mathbf{0}, \sigma_{\mathbf{y}}^2 \mathbf{I})$, where $\sigma_{\mathbf{y}} > 0$ denotes the noise level. Exactly solving these inverse problems is challenging due to their ill-posedness in many settings, requiring advanced techniques to achieve accurate solutions.

Deep learning techniques have recently been utilized to form priors to aid in solving these problems (Ravishankar et al., 2019; Lempitsky et al., 2018). One framework that has shown significant potential is the use of generative models, particularly diffusion models (DMs) (Ho et al., 2020). Given a training dataset, DMs are trained to learn the underlying distribution $p(\mathbf{x})$.

During inference, DMs enable sampling from this learned distribution through an iterative procedure (Song et al., 2021b). When employed to solving inverse problems, DM-based IP solvers often modify the reverse sampling steps to allow sampling from the measurements-conditioned distribution $p(\mathbf{x}|\mathbf{y})$ (Chung et al., 2023b; 2022; Cardoso et al., 2024). These modifications typically rely on approximations that may not be suitable for all tasks and settings, and in addition to generally requiring many sampling iterations, often suffer from errors accumulated during early diffusion sampling steps (Zhang et al., 2025).

In most DM-based IP solvers, these approximations are designed to enforce standard measurement consistency on

the estimated image (or posterior mean) at every reverse sampling iteration, as in (Chung et al., 2023b), and may also include resampling using the forward diffusion process (which we refer to as forward diffusion consistency), such as in (Lugmayr et al., 2022; Song et al., 2023a).

A key bottleneck in DMs is their computational cost, as they are slower than other generative models due to the large number of sampling steps. Although various methods have been proposed to reduce sampling frequency (e.g., (Song et al., 2023c)), these improvements have yet to be fully realized for DMs applied to IPs. Most existing methods still require dense sampling, which continues to pose speed challenges. Based on the aforementioned discussion, in this paper, we make the following contributions.

- **Formulating three conditions for DM-based IP solvers**: By identifying key issues in accelerating DMs for IPs, we present a systematic formulation of three conditions where the goal is to (*i*) improve the accuracy of each intermediate reconstruction and (*ii*) maintain a diffusion trajectory.
- **Introducing the step-wise Backward consistency**: Based on the pre-trained network regularization, we propose the step-wise backward diffusion consistency by leveraging the implicit bias of the pre-trained network at each iteration.
- **Presenting a new optimization-based sampler**: We present an optimization-based sampler that allows for arbitrary sampling step sizes. We refer to our method as **S**tep-w**i**se **T**riple-**Co**nsistent Sa**m**pling (SITCOM). The formulation, initialization, and regularization of the optimization problem along with the preceding steps ensure that all consistencies/conditions are enforced with minimal deviation at every sampling iteration.
- **Extensive Evaluation**: We evaluate SITCOM on one medical image reconstruction task (MRI) and 8 image restoration tasks (5 linear and 3 non-linear). Compared to leading baselines, our approach consistently achieves either state-of-the-art or highly competitive quantitative results, while also reducing the number of sampling steps and, consequently, the computational time.

## 2. Background: DMs & Their Usage for Solving IPs

In this section, we discuss preliminaries on Diffusion Models (DMs) and their usage in solving IPs.

Pre-trained DMs generate images by applying a pre-defined iterative denoising process (Ho et al., 2020). In the Variance-Preserving Stochastic Differentiable Equations (SDEs) setting (Song et al., 2021b;a), DMs are formulated using the forward process and the reverse process

$$dx_t = -\frac{\beta_t}{2}x_t dt + \sqrt{\beta_t}dw \, , \qquad (1a)$$

$$dx_t = -\beta_t \left[ \frac{1}{2}x_t + \nabla_{x_t} \log p_t(x_t) \right] dt + \sqrt{\beta_t}d\bar{w} \, , \quad (1b)$$

where $\beta : \{0, \dots, T\} \to (0, 1)$ is a pre-defined function that controls the amount of additive perturbations at time $t$, $w$ (resp. $\bar{w}$) is the forward (resp. reverse) Weiner process (Anderson, 1982), $p_t(x_t)$ is the distribution of $x_t$ at time $t$, and $\nabla_{x_t} \log p_t(x_t)$ is the score function that is replaced by a neural network (typically a time-encoded U-Net (Ronneberger et al., 2015)) $s : \mathbb{R}^n \times \{0, \dots, T\} \to \mathbb{R}^n$, parameterized by $\theta$. In practice, given the score function $s_\theta$, the SDEs can be discretized as in (2) where $\eta_t, \eta_{t-1} \sim \mathcal{N}(0, I)$.

$$x_t = \sqrt{1 - \beta_t}x_{t-1} + \sqrt{\beta_t}\eta_{t-1} \, , \qquad (2a)$$

$$x_{t-1} = \frac{1}{\sqrt{1 - \beta_t}} \left[ x_t + \beta_t s_\theta(x_t, t) \right] + \sqrt{\beta_t}\eta_t \, . \quad (2b)$$

When employed to solve inverse problems, the score function in (1) is replaced by a conditional score function which, by Bayes' rule, is

$$\nabla_{x_t} \log p_t(x_t|y) = \nabla_{x_t} \log p_t(x_t) + \nabla_{x_t} \log p_t(y|x_t) \, .$$

Solving the SDEs in (1) with the conditional score is referred to as *posterior sampling* (Chung et al., 2023b). As there doesn't exist a closed-form expression for the term $\nabla_{x_t} \log p_t(y|x_t)$ (which is referred to as the measurements matching term in (Daras et al., 2024)), previous works have explored different approaches, which we will briefly discuss below. We refer the reader to the recent survey in (Daras et al., 2024) for an overview on DM-based methods for solving IPs.

A well-known method is Diffusion Posterior Sampling (DPS) (Chung et al., 2023b), which uses the approximation $p(y|x_t) \approx p(y|\hat{x}_0)$ where $\hat{x}_0(x_t)$ (or simply $\hat{x}_0$) is the estimated image at time $t$ as a function of the pre-trained model and $x_t$ (Tweedie's formula (Vincent, 2011)), given as

$$\hat{x}_0(x_t) = \frac{1}{\sqrt{\bar{\alpha}_t}} \left[ x_t - \sqrt{1 - \bar{\alpha}_t}\epsilon_\theta(x_t, t) \right] \qquad (3)$$

$$=: f(x_t; t, \epsilon_\theta) \, ,$$

where

$$\epsilon_\theta(x_t, t) = -\sqrt{1 - \bar{\alpha}_t}s_\theta(x_t, t) \, ,$$

output the noise in $x_t$ (Luo, 2022), and $\bar{\alpha}_t = \prod_{j=1}^{t} \alpha_j$ with $\alpha_t = 1 - \beta_t$. We call the function $f$, defined in (3), as **"Tweedie-network denoiser"** (also termed as 'posterior mean predictor' in (Chen et al., 2024)).

Tweedie's formula, like in our method, is also adopted in other DM-based IP solvers such as (Rout et al., 2023; Chung

et al., 2023c; Wang et al., 2022). The drawback of most of these methods is that they typically require a large number of sampling steps.

The work in ReSample (Song et al., 2023a), solves an optimization problem on the estimated posterior mean in the latent space to enforce a step-wise measurement consistency, requiring many sampling and optimization steps.

The work in (Mardani et al., 2024) introduced RED-Diff, a variational Bayesian method that fits a Gaussian distribution to the posterior distribution of the clean image conditional on the measurements. RED-Diff solves also an optimization problem using stochastic gradient descent (SGD) to minimize a data-fitting term while maximizing the likelihood of the reconstructed image under the denoising diffusion prior (as a regularizer). However, the SGD process requires multiple iterations, each involving evaluations of the pre-trained DM on a different noisy image at some randomly selected time. While RED-diff reduces the run-time, their qualitative results are not highly competitive on some tasks in addition to requiring an external pre-trained denoiser.

Recently, Decoupling Consistency with Diffusion Purification (DCDP) (Li et al., 2024) proposed separating diffusion sampling steps from measurement consistency by using DMs as diffusion purifiers (Nie et al., 2022; Alkhouri et al., 2024b), with the goal of reducing the run-time. However, for every task, DCDP requires tuning the number of forward diffusion steps for purification for each sampling step. Shortly after, Decoupled Annealing Posterior Sampling (DAPS) (Zhang et al., 2025) introduced another decoupled approach, incorporating gradient descent noise annealing via Langevin dynamics. DAPS, similar to DPS, also requires a large number of sampling and optimization steps.

## 3. Step-wise Triple-Consistent Sampling

### 3.1. Motivation: Addressing the Challenges in Applying DMs to IPs

Most inverse problems are ill-conditioned and undersampled. DMs, when trained on a dataset that closely resembles the target image, can provide critical information to alleviate ill-conditioning and improve recovery. Despite various previous efforts, a key challenge remains: How to *efficiently* integrate DMs into the framework of inverse problems? We will now elaborate on this challenge in detail.

The standard reverse sampling procedure in DMs consists of applying the backward discrete steps in (2) for $t \in \{T, T-1, \dots, 1\}$, forming the standard diffusion trajectory for which $\mathbf{x}_0$ is the generated image[1]. To incorporate

---

[1]Diffusion trajectory refers to the path that leads to an in-distribution image, where the distribution is the one learned by the DM from the training set.

the measurements $\mathbf{y}$ into these steps, a common approach adopted in previous works that demonstrate superior performance (e.g., (Song et al., 2023a; Zhang et al., 2025; Li et al., 2024)) is to encode $\hat{\mathbf{x}}_0$, computed via (3), in the following optimization problem:

$$\hat{\mathbf{x}}_0'(\mathbf{x}_t) = \arg \min_{\mathbf{x}} \|\mathcal{A}(\mathbf{x}) - \mathbf{y}\|_2^2 + \lambda \|\mathbf{x} - \hat{\mathbf{x}}_0(\mathbf{x}_t)\|_2^2 \,, \quad (4)$$

where $\lambda \in \mathbb{R}_+$ is a regularization parameter. The $\hat{\mathbf{x}}_0'(\mathbf{x}_t)$ obtained from (4) is close to $\hat{\mathbf{x}}_0(\mathbf{x}_t)$ while also remaining consistent with the measurements. When using $\hat{\mathbf{x}}_0'(\mathbf{x}_t)$ to sample $\mathbf{x}_{t-1}$, the second formula in (2) can be rewritten as in (5), where the derivation is provided in Appendix A.

$$\mathbf{x}_{t-1} = \frac{\sqrt{\alpha_t}(1 - \bar{\alpha}_{t-1})}{1 - \bar{\alpha}_t}\mathbf{x}_t + \frac{\sqrt{\bar{\alpha}_{t-1}}\beta_t}{1 - \bar{\alpha}_t}\hat{\mathbf{x}}_0(\mathbf{x}_t) \quad (5)$$
$$+ \sqrt{\beta_t}\boldsymbol{\eta}_t \,.$$

By substituting $\hat{\mathbf{x}}_0(\mathbf{x}_t)$ in the second term of (5) with the measurement-consistent $\hat{\mathbf{x}}_0'(\mathbf{x}_t)$, the modified sampling formula becomes:

$$\mathbf{x}_{t-1} = \frac{\sqrt{\alpha_t}(1 - \bar{\alpha}_{t-1})}{1 - \bar{\alpha}_t}\mathbf{x}_t + \frac{\sqrt{\bar{\alpha}_{t-1}}\beta_t}{1 - \bar{\alpha}_t}\hat{\mathbf{x}}_0'(\mathbf{x}_t) \quad (6)$$
$$+ \sqrt{\beta_t}\boldsymbol{\eta}_t \,.$$

While this approach effectively ensures data consistency at each step, it inevitably causes $\hat{\mathbf{x}}_0'$ to deviate from the diffusion trajectory, leading to two **major issues**:

**(I1)**: The image $\hat{\mathbf{x}}_0(\mathbf{x}_t)$, initially constructed through Tweedie's formula, usually appears quite natural (e.g., columns 3 to 5 of Figure 1); however, the modified version, $\hat{\mathbf{x}}_0'(\mathbf{x}_t)$, is likely to exhibit severe artifacts (e.g., columns 6 to 8 of Figure 1).

**(I2)**: Since the DM network, $\boldsymbol{\epsilon}_\theta$, is trained via minimizing the objective function (denoising score matching (Vincent, 2011))

$$\mathbb{E}_{\mathbf{x}_0 \sim p(\mathbf{x}_0), \boldsymbol{\epsilon} \sim \mathcal{N}(0, \mathbf{I})} \left\| \boldsymbol{\epsilon} - \boldsymbol{\epsilon}_\theta(\sqrt{\bar{\alpha}_t}\mathbf{x}_0 + \sqrt{1 - \bar{\alpha}_t}\boldsymbol{\epsilon}, t) \right\|_2^2 \,,$$

on a finite dataset, it performs best on noisy images lying in the high-density regions of the *training distribution* $\mathcal{N}(\mathbf{x}_t; \sqrt{\bar{\alpha}_t}\mathbf{x}_0, (1 - \bar{\alpha}_t)\mathbf{I})$ . We define an algorithm as **forward-consistent** if it likely applies $\boldsymbol{\epsilon}_\theta$ only to in-distribution inputs (i.e., those from the same distribution used for training). For example, if the forward diffusion used to train $\boldsymbol{\epsilon}_\theta$ adds Gaussian noise, the in-distribution input to $\boldsymbol{\epsilon}_\theta$ should ideally be sampled from a Gaussian with specific parameters. If Poisson noise is used in the forward process, inputs drawn from suitable Poisson distributions are more likely to fall within the well-trained region of the network. In summary, forward consistency requires that inputs to $\boldsymbol{\epsilon}_\theta$ during sampling align with the forward process. While the $\mathbf{x}_{t-1}$ generated from (5) is forward-consistent by

design, the one generated from the modified formula (6) is not. Therefore, in the latter case, the DM network, $\epsilon_\theta$, may be applied to many out-of-distribution inputs, leading to degraded performance.

We pause to verify our claimed Issue **(I1)** through a box-inpainting experiment. Columns 6 to 8 of Figure 1 show $\hat{\mathbf{x}}_0'(\mathbf{x}_t)$ at various $t'$. The results clearly demonstrate successful enforcement of data consistency, as the region outside the box aligns with the original image. However, this enforcement compromises the natural appearance of the image, introducing significant artifacts in the reconstructed area inside the box. Details about the setting of the results in Figure 1 are given in Appendix C.1.2.

Issue **(I2)** was previously observed in (Lugmayr et al., 2022), which proposed a remedy known as '*resampling*'. In this approach, the sampling formula in (6) is replaced by

$$\mathbf{x}_{t-1} = \sqrt{\bar{\alpha}_{t-1}}\hat{\mathbf{x}}_0 + \sqrt{1 - \bar{\alpha}_{t-1}}\boldsymbol{\eta}_t .\qquad(7)$$

Provided $\hat{\mathbf{x}}_0$ is close to the ground truth $\mathbf{x}_0$, $\mathbf{x}_{t-1}$ generated this way will stay in-distribution with high probability. For a more detailed explanation of the rationale behind this remedy, we refer the reader to (Lugmayr et al., 2022). This method has since been adopted by subsequent works, such as (Song et al., 2023a; Zhang et al., 2025), and we will also employ it to address **(I2)**.

## 3.2. Network-Regularized Backward Diffusion Consistency

Previous studies, such as (Song et al., 2023a; Zhang et al., 2025), mitigate issue **(I1)** by using a large number of sampling steps, which inevitably increases the computational burden. In contrast, this paper proposes employing a *network regularization* to resolve issue **(I1)**. This approach not only accelerates convergence but also enhances reconstruction quality. Let's first clarify the underlying intuition.

It is widely observed that the U-Net architecture or trained transformers exhibit an effective image bias (Ulyanov et al., 2018; Liang et al., 2025; Hatamizadeh et al., 2025). From columns 3 to 5 of Figure 1, we observe that without enforcing data consistency, the reconstructed $\hat{\mathbf{x}}_0$, derived directly from Tweedie-network denoiser $f(\mathbf{x}_t; t, \epsilon_\theta)$ for each time $t$, exhibits natural textures. This indicates that the reconstruction using the combination of Tweedie's formula and the DM network has a natural regularizing effect on the image.

By definition, the output of $f(\mathbf{x}_t; t, \epsilon_\theta)$ in (3) represents the *denoised* version of $\mathbf{x}_t$ at time $t$ using the Tweedie's formula and the DM denoiser $\epsilon_\theta$. Due to the implicit bias of $\epsilon_\theta$, this denoised image tends to align with the clean image manifold, even if $\mathbf{x}_t$ does not correspond to a training image, as shown in columns 3 to 5 of Figure 1. We refer to this regularization effect of $f(\mathbf{x}_t; t, \epsilon_\theta)$, which arises from

network bias, as "**network regularization**".

By employing network regularization, we can address **(I1)** by ensuring that the data-consistent $\hat{\mathbf{x}}_0'$ is also network-consistent. We refer the latter condition as **Backward Consistency** and define it formally as follows.

**Definition 3.1** (Backward Consistency)**.** We say a reconstruction $\hat{\mathbf{x}}_0'$ is backward-consistent with posterior mean predictor $f(\,\cdot\,; t, \epsilon_\theta)$ at time $t$ if it can be expressed as $\hat{\mathbf{x}}_0' = f(\mathbf{v}_t; t, \epsilon_\theta)$ with some $\mathbf{v}_t$. In other words, backward consistency requires $\hat{\mathbf{x}}_0'$ to be an output of $f$ at time $t$.

The subset of images that are in the range of function $f$ (i.e., backward-consistent) is denoted by $\mathcal{C}_t$ and is defined as

$$\mathcal{C}_t := \{f(\mathbf{v}_t; t, \epsilon_\theta) : \mathbf{v}_t \in \mathbb{R}^n\} .\qquad(8)$$

Enforcing $\hat{\mathbf{x}}_0'$ to be both measurement- and backward-consistent involves solving the following optimization problem:

$$\hat{\mathbf{x}}_0', \hat{\mathbf{v}}_t := \underset{\mathbf{v}_t', \mathbf{x}_0'}{\operatorname{argmin}} \left\{ \|\mathcal{A}(\mathbf{x}_0') - \mathbf{y}\|_2^2 \right.\qquad(9)$$
$$\text{subject to}\quad \mathbf{x}_0' = f(\mathbf{v}_t'; t, \epsilon_\theta) \big\} .$$

However, (9) may violate forward consistency, as $\hat{\mathbf{v}}_t$ could possibly be far from $\mathbf{x}_t$. Therefore, we propose adding a regularization term, for which (9) becomes

$$\hat{\mathbf{x}}_0', \hat{\mathbf{v}}_t := \underset{\mathbf{v}_t', \mathbf{x}_0'}{\operatorname{argmin}} \left\{ \|\mathcal{A}(\mathbf{x}_0') - \mathbf{y}\|_2^2 + \lambda\|\mathbf{x}_t - \mathbf{v}_t'\|_2^2 \right.\quad(10)$$
$$\text{subject to}\quad \mathbf{x}_0' = f(\mathbf{v}_t'; t, \epsilon_\theta) \big\} .$$

During the reverse sampling process, at each time $t$, with the given $\mathbf{x}_t$, we seek a $\mathbf{v}_t'$ in the nearby region (i.e., $\|\mathbf{x}_t - \mathbf{v}_t'\|_2^2$ is small), such that $\mathbf{v}_t'$ can be denoised by $f$ to produce a clean image $\mathbf{x}_0'$ (i.e., $\mathbf{x}_0' = f(\mathbf{v}_t'; t, \epsilon_\theta)$), which is also consistent with the measurements $\mathbf{y}$ (i.e., $\|\mathcal{A}(\mathbf{x}_0') - \mathbf{y}\|_2^2$ is small). We need to identify such a $\mathbf{v}_t'$ because $\mathbf{x}_t$ itself cannot be directly denoised by $f$ to yield an image consistent with the measurements. By substituting the constraint into the objective function, the optimization problem in (10) is reduced to

$$\hat{\mathbf{v}}_t := \underset{\mathbf{v}_t'}{\operatorname{argmin}} \left\{ \|\mathcal{A}(f(\mathbf{v}_t'; t, \epsilon_\theta)) - \mathbf{y}\|_2^2 + \right.\quad(11)$$
$$\lambda\|\mathbf{x}_t - \mathbf{v}_t'\|_2^2 \big\} ,$$

with $\hat{\mathbf{x}}_0' = f(\hat{\mathbf{v}}_t; t, \epsilon_\theta)$. The benefit of the considered backward consistency constraint is shown in columns 9 to 11 of Figure 1. After obtaining $\hat{\mathbf{x}}_0'$, the resampling formula in (7) is used to obtain $\mathbf{x}_{t-1}$.

### 3.2.1. RELATION TO GENERATIVE PRIORS

The use of network regularization to define step-wise backward consistency is inspired by Compressed Sensing with

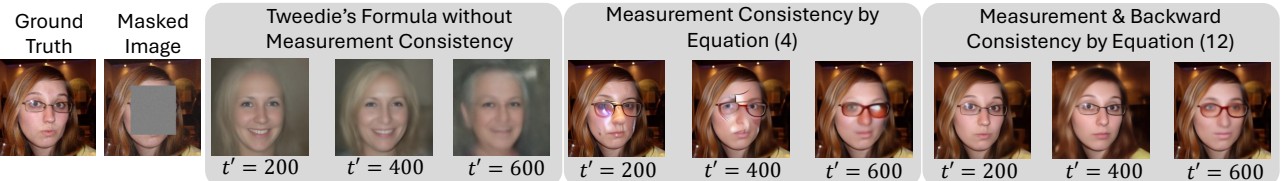

Figure 1: Effects of enforcing backward-consistency in box-inpainting: Results of using Tweedie's formula without measurement consistency (columns 3 to 5), enforcing measurement-consistency via (4) (columns 6 to 8), and enforcing both measurement-consistency and backward-consistency via (11) (columns 9 to 11) at different time steps $t'$. Experimental details are given in Appendix C.1.2.

generative models (CSGM) (Bora et al., 2017). In CSGM, given a pre-trained generative model $g_\phi$ with $\phi$ as its weights, it can regularize the reconstruction of inverse problems by solving the following optimization problem:

$$\hat{\mathbf{z}} = \underset{\mathbf{z}}{\arg\min} \|\mathcal{A}g_\phi(\mathbf{z}) - \mathbf{y}\|_2^2 \,, \qquad (12)$$

and $\mathbf{x} = g_\phi(\hat{\mathbf{z}})$ is the reconstructed image. Here, $\mathbf{z}$ is the input of the pre-trained network and the optimization variable. In this setup, the reconstruction $\mathbf{x}$ is constrained to be the output of the network $g_\phi$. Similarly, in Definition 3.1, $\hat{\mathbf{x}}_0'$ is required to be the output of the posterior mean estimator $f$, which is defined by the network $\boldsymbol{\epsilon}_\theta$.

### 3.3. Triple Consistency Conditions

We now summarize the three key conditions that apply at each sampling step.

**C1** **Measurement Consistency**: The reconstruction $\hat{\mathbf{x}}_0'$ is consistent with the measurements. This means that $\mathcal{A}(\hat{\mathbf{x}}_0') \approx \mathbf{y}$.

**C2** **Backward Consistency**: The reconstruction $\hat{\mathbf{x}}_0'$ is a denoised image produced by the Tweedie-network denoiser $f$. More generally, we define the backward consistency to include any form of DM network regularization (e.g., using the DM probability-flow (PF) ODE (Karras et al., 2022) as in Appendix F) applied to $\hat{\mathbf{x}}_0'$.

**C3** **Forward Consistency**: The pre-trained DM network $\boldsymbol{\epsilon}_\theta$ is provided with in-distribution inputs with high probability. To ensure this, we apply the resampling formula in (7) and enforce that $\hat{\mathbf{v}}_t$ remains close to $\mathbf{x}_t$.

We note that the three considered consistencies are step-wise, meaning they are enforced at every sampling step. This approach contrasts with enforcing these consistencies solely on the final reconstruction at $t = 0$, which represents a significantly weaker requirement.

**C1**-**C3** aim to ensure that all intermediate reconstructions $\hat{\mathbf{x}}_0'(\cdot) = f(\,\cdot\,; t, \boldsymbol{\epsilon}_\theta)$ (with $t > 0$) are as accurate as possible, allowing us to effectively reduce the number of sampling steps. Previous works, such as (Song et al., 2023a; Zhang et al., 2025), enforce measurement consistency by applying $\mathcal{A}(\hat{\mathbf{x}}_0) = \mathbf{y}$ exactly, whereas DPS (Chung et al., 2023b) does not ensure forward and a step-wise backward consis-

tencies along the diffusion trajectory.

### 3.4. The Proposed Sampler

Given $\mathbf{x}_t, \boldsymbol{\epsilon}_\theta$, and towards satisfying the above conditions, our method, at sampling time $t$, consists of the following three steps:

$$\hat{\mathbf{v}}_t := \underset{\mathbf{v}_t'}{\arg\min} \quad \left\| \mathcal{A}\left( \tfrac{1}{\sqrt{\bar{\alpha}_t}} \left[\mathbf{v}_t' - \sqrt{1 - \bar{\alpha}_t}\, \boldsymbol{\epsilon}_\theta(\mathbf{v}_t', t)\right] \right) - \mathbf{y} \right\|_2^2$$
$$+ \lambda \|\mathbf{x}_t - \mathbf{v}_t'\|_2^2 \qquad (\text{S}_1)$$

$$\hat{\mathbf{x}}_0' = f(\hat{\mathbf{v}}_t; t, \boldsymbol{\epsilon}_\theta) \equiv \tfrac{1}{\sqrt{\bar{\alpha}_t}}\left[\hat{\mathbf{v}}_t - \sqrt{1 - \bar{\alpha}_t}\, \boldsymbol{\epsilon}_\theta(\hat{\mathbf{v}}_t, t)\right] \quad (\text{S}_2)$$

$$\mathbf{x}_{t-1} = \sqrt{\bar{\alpha}_{t-1}}\hat{\mathbf{x}}_0' + \sqrt{1 - \bar{\alpha}_{t-1}}\boldsymbol{\eta}_t \,. \qquad (\text{S}_3)$$

In (S₁), the argument of the forward operator is $f(\mathbf{v}'_t; t, \boldsymbol{\epsilon}_\theta) = \tfrac{1}{\sqrt{\bar{\alpha}_t}}\left[\mathbf{v}_t' - \sqrt{1 - \bar{\alpha}_t}\, \boldsymbol{\epsilon}_\theta(\mathbf{v}_t', t)\right]$ which is different from (S₂).

The minimization in the first step optimizes over the input $\mathbf{v}_t'$ of the pre-trained diffusion model at time $t$, where the first term of the objective enforces measurement consistency for the posterior mean estimated image, satisfying condition **C1**. The second term serves as a regularization term, implicitly enforcing closeness between $\hat{\mathbf{v}}_t$ and $\mathbf{x}_t$ (i.e., condition **C3**), with $\lambda > 0$ acting as the regularization parameter.

The argument of the forward operator in (S₁) and the second step in (S₂) enforce that $\hat{\mathbf{v}}_t$ and $\hat{\mathbf{x}}_0'$, respectively, maintain the diffusion trajectory through obeying Tweedie's formula, thereby satisfying the backward consistency condition, **C2**.

After obtaining the measurement-consistent estimate, $\hat{\mathbf{x}}_0'$, as given in (S₂), it must be mapped back to time $t - 1$ to generate $\mathbf{x}_{t-1}$. This is achieved through the forward diffusion step in (S₃) as outlined in the forward consistency condition, **C3**. A diagram of the SITCOM procedure is provided in Figure 2 (left).

**Remark 1.** Obtaining the estimated image at time $0$ given some $\mathbf{x}_t$ using the standard DM PF-ODE (Karras et al., 2022) is more accurate compared to the one-step Tweedie's formula. However, since PF-ODE is an iterative procedure,

**Algorithm 1** **S**tep-w**i**se **T**riple-**Co**nsistent Sa**m**pling (**SITCOM**).

**Input**: Measurements $\mathbf{y}$, forward operator $\mathcal{A}(\cdot)$, pre-trained DM $\boldsymbol{\epsilon}_\theta(\cdot, \cdot)$, number of diffusion steps $N$, DM noise schedule $\bar{\alpha}_i$ for $i \in \{1, \ldots, N\}$, number of gradient updates $K$, stopping criterion $\delta$, learning rate $\gamma$, and regularization parameter $\lambda$.
 **Output**: Restored image $\hat{\mathbf{x}}$.

**Initialization**: $\mathbf{x}_N \sim \mathcal{N}(\mathbf{0}, \mathbf{I})$, $\Delta t = \lfloor \frac{T}{N} \rfloor$

1: **For each** $i \in \{N, N-1, \ldots, 1\}$. (Reducing diffusion sampling steps)

2:   **Initialize** $\mathbf{v}_i^{(0)} \leftarrow \mathbf{x}_i$. (Initialization to ensure Closeness: **C3** )

3:   **For each** $k \in \{1, \ldots, K\}$. (Gradient used in Adam to achieve measurement & backward consistency: **C1**, **C2**)

4:     $\mathbf{v}_i^{(k)} = \mathbf{v}_i^{(k-1)} - \gamma \nabla_{\mathbf{v}_i} \left[ \left\| \mathcal{A}\left( \frac{1}{\sqrt{\bar{\alpha}_i}} \left[ \mathbf{v}_i - \sqrt{1 - \bar{\alpha}_i}\, \boldsymbol{\epsilon}_\theta(\mathbf{v}_i, i\Delta t) \right] \right) - \mathbf{y} \right\|_2^2 + \lambda \| \mathbf{x}_i - \mathbf{v}_i \|_2^2 \right] \Big|_{\mathbf{v}_i = \mathbf{v}_i^{(k-1)}}$.

5:       **If** $\left\| \mathcal{A}\left( \frac{1}{\sqrt{\bar{\alpha}_i}} \left[ \mathbf{v}_i^{(k)} - \sqrt{1 - \bar{\alpha}_i}\, \boldsymbol{\epsilon}_\theta(\mathbf{v}_i^{(k)}, i\Delta t) \right] \right) - \mathbf{y} \right\|_2^2 < \delta^2$ . (Stopping criterion)

6:         **Break** the **For** loop in step 3. (Preventing noise overfitting)

7:   **Assign** $\hat{\mathbf{v}}_i \leftarrow \mathbf{v}_i^{(k)}$. (Backward diffusion consistency of $\hat{\mathbf{v}}_i$: **C2**)

8:   **Obtain** $\hat{\mathbf{x}}_0' = f(\hat{\mathbf{v}}_i; t, \theta) = \frac{1}{\sqrt{\bar{\alpha}_i}} \left[ \hat{\mathbf{v}}_i - \sqrt{1 - \bar{\alpha}_i}\, \boldsymbol{\epsilon}_\theta(\hat{\mathbf{v}}_i, i\Delta t) \right]$. (Backward consistency of $\hat{\mathbf{x}}_0'$: **C2**)

9:   **Obtain** $\mathbf{x}_{i-1} = \sqrt{\bar{\alpha}_{i-1}} \hat{\mathbf{x}}_0' + \sqrt{1 - \bar{\alpha}_{i-1}} \boldsymbol{\eta}_i$, $\boldsymbol{\eta}_i \sim \mathcal{N}(\mathbf{0}, \mathbf{I})$ . (Forward diffusion consistency: **C3**)

10: **Restored image:** $\hat{\mathbf{x}} = \mathbf{x}_0$.

it requires more computational time. In SITCOM, PF-ODE could replace Tweedie's formula in ($\text{S}_2$). In the main body of the paper, we chose not to use it, as this would increase the run time, and our empirical results are already highly competitive using Tweedie's formula. To show this trade off, we refer the reader to the study in Appendix F.

A conceptual illustration of SITCOM is shown in Figure 2 (*right*). The DM generative manifold, $\mathcal{M}_t$, is defined as the set of all $\mathbf{x}_t$ sampled from

$$q(\mathbf{x}_t | \mathbf{x}_0) = \mathcal{N}(\mathbf{x}_t; \sqrt{\bar{\alpha}_t} \mathbf{x}_0, (1 - \bar{\alpha}_t)\mathbf{I}) \ ,$$

and $\mathbf{x}_0 \sim p_0(\mathbf{x})$. This set coincides with the entire space $\mathbb{R}^n$ equipped with the probability measure induced by the distribution of $\mathbf{x}_t$, which we denote as $\mathcal{P}_t$. In Figure 2 (*right*), the variation of color around each $\mathcal{M}_t$ indicates the concentration of the measure $\mathcal{P}_t$, with darker colors representing higher concentration.

SITCOM's Step (1) and Step (2) enforce measurement consistency and backward consistency, thus map $\mathbf{x}_t$ to $\hat{\mathbf{x}}_0' = f(\hat{\mathbf{v}}_t; t, \boldsymbol{\epsilon}_\theta)$ which lies within the intersection of (*i*) measurement-consistent set $\{\hat{\mathbf{x}}_0' : \mathcal{A}(\hat{\mathbf{x}}_0') \approx \mathbf{y}\}$ (the shaded black line) and (*ii*) the backward-consistent set $\mathcal{C}_t$ (the yellow ellipsoid). Subsequently, $\mathbf{x}_{t-1}$ is generated by inserting $\hat{\mathbf{x}}_0'$ into the resampling formula, which enforces the forward consistency.

**Handling Measurement Noise:**   To avoid the case where the first term of the objective in ($\text{S}_1$) reaches small values yielding noise overfitting (i.e., when additive Gaussian measurement noise is considered, $\sigma_{\mathbf{y}} > 0$), we propose refraining from enforcing strict measurement fitting $\mathcal{A}(\mathbf{x}) = \mathbf{y}$.

Instead, we use the stopping criterion

$$\left\| \mathcal{A}\left( \frac{1}{\sqrt{\bar{\alpha}_t}} \left[ \mathbf{v}_t' - \sqrt{1 - \bar{\alpha}_t}\, \boldsymbol{\epsilon}_\theta(\mathbf{v}_t', t) \right] \right) - \mathbf{y} \right\|_2^2 < \delta^2 \ ,$$

where $\delta \in \mathbb{R}_+$ is a hyper-parameter that indicates tolerance for noise and helps prevent overfitting. This is equivalent to enforcing an $\ell_2$ constraint. In our experiments, similar to DAPS (Zhang et al., 2025) and PGDM (Song et al., 2023b), we use $\delta$ slightly larger than the actual level of noise in the measurements, i.e., $\delta > \sigma_{\mathbf{y}} \sqrt{m}$.

**Remark 2.** Setting the stopping criterion based on the noise level in the measurements may not be practical, as noise estimators may be inaccurate. However, in Appendix J.2.1, we empirically show that SITCOM is not sensitive to the choice of this threshold by demonstrating that even when the stopping criterion is set higher or lower than the actual measurement noise level, the performance remains largely unaffected.

### 3.5. SITCOM with Arbitrary Stepsizes

Here, we explain how to apply SITCOM with a large step-size and present the final algorithm. The DM is trained with $T$ diffusion steps. Given that our method is designed to satisfy the three consistencies, SITCOM requires $N \ll T$ sampling iterations, using a step size of $\Delta t := \lfloor \frac{T}{N} \rfloor$. Thus, we introduce the index $i$ instead of $t$ with a relation $t_i = i\Delta t$.

The procedure of SITCOM is outlined in Algorithm 1. As inputs, SITCOM takes $\mathbf{y}$, $\mathcal{A}(\cdot)$, $\boldsymbol{\epsilon}_\theta$, the number of sampling steps $N$, $\bar{\alpha}_i$ for all $i \in \{1, \ldots, N\}$, the number of optimization steps $K$ per sampling step, stopping criteria $\delta$, and the learning rate $\gamma$.

Starting with initializing $\mathbf{v}_i^{(0)}$ as $\mathbf{x}_i$ (satisfying condition **C3**), lines 3 through 6 correspond to the gradient updates

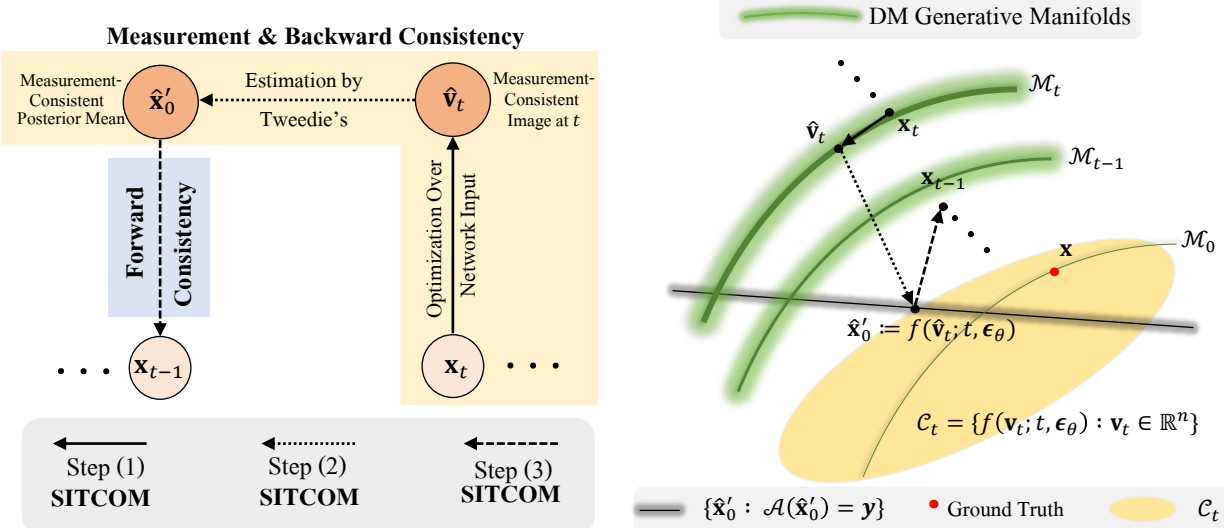

Figure 2: Illustrative diagram of the proposed procedure in SITCOM (*left*). Conceptual illustration of SITCOM, where $\mathcal{M}_t$ is the DM generative manifold at time $t$ and $\mathcal{C}_t$ is the subset of images that are backward-consistent (*right*).

for solving the optimization problem in the first step of SIT-COM, i.e., $(\text{S}_1)^2$. In lines 5 and 6, the stopping criterion is applied to prevent strict data fidelity (avoiding noise overfitting). Following the gradient updates in the inner loop, $\hat{\mathbf{v}}_i$ is obtained in line 7, which is then used in line 8 to obtain $\hat{\mathbf{x}}'_0$ as specified in $(\text{S}_2)$, satisfying condition **C2**. Note that line 8 requires no additional computation, as the $\hat{\mathbf{x}}'_0$ calculated here was already obtained while checking the stopping condition in line 6. After obtaining the double-consistent $\hat{\mathbf{x}}'_0$, the resampling is applied to map the image back to time $t-1$ while ensuring $\mathbf{x}_{t-1}$ to be in-distribution, as indicated in line 9 of the algorithm. In the next iteration, the requirement that $\hat{\mathbf{v}}_{t-1}$ is close to $\mathbf{x}_{t-1}$ ensures that the input $\hat{\mathbf{v}}_{t-1}$ to the DM network, $\epsilon_\theta$, is also in-distribution, thus satisfying the forward-consistency (condition **C3**).

The computational requirements of SITCOM are determined by (*i*) the number of sampling steps $N$ and (*ii*) the number of gradient steps $K$ required for each sampling iteration. Given the proposed stopping criterion, this results in at most $NK$ Number of Function Evaluations (NFEs) of the pre-trained model (forward pass), $NK$ backward passes through the pre-trained model, and $NK$ applications each for the forward operator and its adjoint to solve the optimization problem in $(\text{S}_1)$. With early stopping, the computational cost is lower. For example, for a linear operator $\mathcal{A}$ with dimensions $m \times n$, the cost of applying it (or its adjoint) to a vector is $\mathcal{O}(mn)$. For a network with width $M$ and depth $L$, the cost for making a forward pass is $\mathcal{O}(LM^2)$.

The gradients are computed w.r.t. the input of the DM

---

[2]Although lines 3 to 6 of Algorithm 1 describe using fixed-step-size gradient descent for $(\text{S}_1)$, we note that, in our experiments, we use the ADAM optimizer (Kingma & Ba, 2015).

network, requiring an additional backward pass. This backward pass has the same computational cost as the forward pass. Consequently, this procedure is significantly more efficient than network training, where the network weights are updated instead of the input.

In Appendix D, we offer high-level insights to enhance the understanding of SITCOM, particularly about where its acceleration comes from. In essence, we interpret SITCOM with $K$ inner iterations as an accelerated variant of DPS, under certain approximations, via ADAM or other precondition GD (Proposition D.1).

### 3.6. Relation to Existing Studies

Both SITCOM and the works in DAPS (Zhang et al., 2025), DCDP (Li et al., 2024), and ReSample (Song et al., 2023a) are optimization-based methods that modify (or decouple) the sampling steps to enforce measurement consistency. DAPS and ReSample involve mapping back to time $t-1$ (as in step 3 of SITCOM) and DCDP uses this step prior to purification. However, there is a major difference between them: The optimization variable in these works is the estimated image at time $t$ (the output of the DM network), whereas in SITCOM, it is the noisy image at time $t$ (the input of the network). This means that these studies enforce **C1** and **C3** (only DAPS and ReSample), but not **C2**.

Previous works, RED-diff (Mardani et al., 2024) and DM-Plug (Wang et al., 2024), also utilize the implicit bias of the pre-trained network. However, they adopt the full diffusion process as a regularizer, applied only once. In contrast, SITCOM uses the neural network as the regularizer *at each iteration* and focuses specifically on reducing the number of sampling steps for a given level of accuracy.

| Task | Method | FFHQ | | | | ImageNet | | | |
|---|---|---|---|---|---|---|---|---|---|
| | | PSNR (↑) | SSIM (↑) | LPIPS (↓) | Run-time (↓) | PSNR (↑) | SSIM (↑) | LPIPS (↓) | Run-time (↓) |
| SR | DPS | $24.44_{\pm0.56}$ | $0.801_{\pm0.032}$ | $0.26_{\pm0.022}$ | $75.60_{\pm15.20}$ | $23.86_{\pm0.34}$ | $0.76_{\pm0.041}$ | $0.357_{\pm0.069}$ | $142.80_{\pm21.20}$ |
| | DAPS | $29.24_{\pm0.42}$ | $0.851_{\pm0.024}$ | $\mathbf{0.135}_{\pm0.039}$ | $74.40_{\pm13.20}$ | $\underline{25.67}_{\pm0.73}$ | $\underline{0.802}_{\pm0.045}$ | $\underline{0.256}_{\pm0.067}$ | $129.60_{\pm17.00}$ |
| | DDNM | $28.02_{\pm0.78}$ | $0.842_{\pm0.034}$ | $0.197_{\pm0.034}$ | $64.20_{\pm25.20}$ | $23.96_{\pm0.89}$ | $0.767_{\pm0.045}$ | $0.475_{\pm0.044}$ | $76.20_{\pm23.00}$ |
| | DMPlug | $\underline{30.18}_{\pm0.67}$ | $\underline{0.862}_{\pm0.056}$ | $0.149_{\pm0.045}$ | $68.2_{\pm21.12}$ | – | – | – | – |
| | RED-diff | – | – | – | – | $24.89_{\pm0.55}$ | $0.789_{\pm0.056}$ | $0.324_{\pm0.042}$ | $\mathbf{18.02}_{\pm4.00}$ |
| | PGDM | – | – | – | – | $25.22_{\pm0.89}$ | $0.798_{\pm0.037}$ | $0.289_{\pm0.04}$ | $\underline{26.2}_{\pm7.20}$ |
| | DCDP | $27.88_{\pm1.34}$ | $0.825_{\pm0.07}$ | $0.211_{\pm0.05}$ | $\mathbf{19.10}_{\pm4.40}$ | $24.12_{\pm1.24}$ | $0.772_{\pm0.032}$ | $0.351_{\pm0.045}$ | $55.00_{\pm11.22}$ |
| | SITCOM (ours) | $\mathbf{30.68}_{\pm1.02}$ | $\mathbf{0.867}_{\pm0.045}$ | $\underline{0.142}_{\pm0.056}$ | $\underline{27.00}_{\pm4.80}$ | $\mathbf{26.35}_{\pm1.21}$ | $\mathbf{0.812}_{\pm0.021}$ | $\mathbf{0.232}_{\pm0.038}$ | $\underline{67.20}_{\pm11.20}$ |
| BIP | DPS | $23.20_{\pm0.89}$ | $0.754_{\pm0.023}$ | $0.196_{\pm0.033}$ | $94.20_{\pm23.00}$ | $19.78_{\pm0.78}$ | $0.691_{\pm0.052}$ | $0.312_{\pm0.025}$ | $136.80_{\pm22.20}$ |
| | DAPS | $24.17_{\pm1.02}$ | $0.787_{\pm0.032}$ | $\underline{0.135}_{\pm0.032}$ | $81.00_{\pm17.00}$ | $21.43_{\pm0.40}$ | $\underline{0.736}_{\pm0.020}$ | $\underline{0.218}_{\pm0.021}$ | $116.40_{\pm35.20}$ |
| | DDNM | $\underline{24.37}_{\pm0.45}$ | $\underline{0.792}_{\pm0.024}$ | $0.232_{\pm0.026}$ | $61.20_{\pm5.92}$ | $\underline{21.64}_{\pm0.66}$ | $0.732_{\pm0.028}$ | $0.319_{\pm0.015}$ | $87.00_{\pm16.20}$ |
| | DCDP | $23.66_{\pm1.67}$ | $0.762_{\pm0.07}$ | $0.144_{\pm0.05}$ | $\mathbf{16.60}_{\pm7.20}$ | $20.45_{\pm1.22}$ | $0.712_{\pm0.07}$ | $0.298_{\pm0.04}$ | $\mathbf{51.62}_{\pm15.00}$ |
| | SITCOM (ours) | $\mathbf{24.68}_{\pm0.78}$ | $\mathbf{0.801}_{\pm0.042}$ | $\mathbf{0.121}_{\pm0.08}$ | $\underline{21.00}_{\pm6.00}$ | $\mathbf{21.88}_{\pm0.92}$ | $\mathbf{0.742}_{\pm0.032}$ | $\mathbf{0.214}_{\pm0.021}$ | $\underline{67.20}_{\pm12.00}$ |
| RIP | DPS | $28.39_{\pm0.82}$ | $0.844_{\pm0.042}$ | $0.194_{\pm0.021}$ | $91.20_{\pm18.00}$ | $24.26_{\pm0.42}$ | $0.772_{\pm0.02}$ | $0.326_{\pm0.034}$ | $136.20_{\pm15.00}$ |
| | DAPS | $\underline{31.02}_{\pm0.45}$ | $\underline{0.902}_{\pm0.015}$ | $\underline{0.098}_{\pm0.017}$ | $93.60_{\pm24.00}$ | $28.44_{\pm0.45}$ | $0.872_{\pm0.024}$ | $\underline{0.135}_{\pm0.052}$ | $128.40_{\pm27.00}$ |
| | DDNM | $29.93_{\pm0.67}$ | $0.889_{\pm0.032}$ | $0.122_{\pm0.056}$ | $87.00_{\pm14.70}$ | $\underline{29.22}_{\pm0.55}$ | $\underline{0.912}_{\pm0.034}$ | $0.191_{\pm0.048}$ | $92.40_{\pm18.20}$ |
| | DMPlug | $31.01_{\pm0.46}$ | $0.899_{\pm0.076}$ | $0.099_{\pm0.037}$ | $72.02_{\pm15.23}$ | – | – | – | – |
| | DCDP | $28.59_{\pm0.95}$ | $0.852_{\pm0.06}$ | $0.202_{\pm0.075}$ | $\mathbf{20.14}_{\pm8.00}$ | $26.22_{\pm1.13}$ | $0.791_{\pm0.06}$ | $0.289_{\pm0.03}$ | $\mathbf{49.50}_{\pm10.40}$ |
| | SITCOM (ours) | $\mathbf{32.05}_{\pm1.02}$ | $\mathbf{0.909}_{\pm0.09}$ | $\mathbf{0.095}_{\pm0.025}$ | $\underline{27.00}_{\pm12.00}$ | $\mathbf{29.60}_{\pm0.78}$ | $\mathbf{0.915}_{\pm0.028}$ | $\mathbf{0.127}_{\pm0.039}$ | $\underline{68.40}_{\pm22.00}$ |
| GDB | DPS | $25.52_{\pm0.78}$ | $0.826_{\pm0.052}$ | $0.211_{\pm0.017}$ | $90.00_{\pm14.45}$ | $21.86_{\pm0.45}$ | $0.772_{\pm0.08}$ | $0.362_{\pm0.034}$ | $153.00_{\pm27.00}$ |
| | DAPS | $29.22_{\pm0.50}$ | $\underline{0.884}_{\pm0.056}$ | $0.164_{\pm0.032}$ | $84.00_{\pm31.20}$ | $26.12_{\pm0.78}$ | $0.832_{\pm0.092}$ | $\underline{0.245}_{\pm0.022}$ | $133.80_{\pm31.20}$ |
| | DDNM | $28.22_{\pm0.52}$ | $0.867_{\pm0.056}$ | $0.216_{\pm0.042}$ | $93.60_{\pm17.00}$ | $\mathbf{28.06}_{\pm0.52}$ | $\mathbf{0.879}_{\pm0.072}$ | $0.278_{\pm0.089}$ | $105.00_{\pm21.80}$ |
| | DCDP | $26.67_{\pm0.78}$ | $0.835_{\pm0.08}$ | $0.196_{\pm0.04}$ | $\mathbf{21.07}_{\pm8.80}$ | $23.24_{\pm1.18}$ | $0.781_{\pm0.06}$ | $0.343_{\pm0.04}$ | $\mathbf{48.25}_{\pm15.80}$ |
| | DMPlug | $\underline{29.79}_{\pm1.02}$ | $0.883_{\pm0.045}$ | $\underline{0.157}_{\pm0.052}$ | $65.44_{\pm22.12}$ | – | – | – | – |
| | SITCOM (ours) | $\mathbf{30.25}_{\pm0.89}$ | $\mathbf{0.892}_{\pm0.032}$ | $\mathbf{0.135}_{\pm0.078}$ | $\underline{27.60}_{\pm8.40}$ | $\underline{27.40}_{\pm0.45}$ | $\underline{0.854}_{\pm0.045}$ | $\mathbf{0.236}_{\pm0.039}$ | $\underline{66.00}_{\pm18.20}$ |
| MDB | DPS | $23.40_{\pm1.42}$ | $0.737_{\pm0.024}$ | $0.270_{\pm0.025}$ | $144.00_{\pm23.00}$ | $21.86_{\pm2.05}$ | $0.724_{\pm0.022}$ | $0.357_{\pm0.032}$ | $153.60_{\pm24.20}$ |
| | RED-diff | – | – | – | – | $27.35_{\pm0.89}$ | $0.842_{\pm0.062}$ | $0.231_{\pm0.045}$ | $\mathbf{19.02}_{\pm8.24}$ |
| | PGDM | – | – | – | – | $27.48_{\pm0.78}$ | $0.848_{\pm0.056}$ | $0.225_{\pm0.052}$ | $\underline{24.1}_{\pm7.45}$ |
| | DAPS | $\underline{29.66}_{\pm0.50}$ | $\underline{0.872}_{\pm0.027}$ | $\underline{0.157}_{\pm0.012}$ | $91.60_{\pm7.20}$ | $27.86_{\pm1.20}$ | $0.862_{\pm0.032}$ | $0.196_{\pm0.021}$ | $118.00_{\pm27.00}$ |
| | SITCOM (ours) | $\mathbf{30.34}_{\pm0.67}$ | $\mathbf{0.902}_{\pm0.037}$ | $\mathbf{0.148}_{\pm0.041}$ | $\underline{30.00}_{\pm7.10}$ | $\mathbf{28.65}_{\pm0.34}$ | $\mathbf{0.876}_{\pm0.021}$ | $\mathbf{0.189}_{\pm0.036}$ | $88.80_{\pm21.00}$ |
| PR | DPS | $17.34_{\pm2.67}$ | $0.67_{\pm0.045}$ | $0.41_{\pm0.08}$ | $90.00_{\pm20.40}$ | $16.82_{\pm1.22}$ | $0.64_{\pm0.08}$ | $0.447_{\pm0.032}$ | $130.20_{\pm14.40}$ |
| | DAPS | $\mathbf{30.97}_{\pm3.12}$ | $\underline{0.908}_{\pm0.041}$ | $\mathbf{0.112}_{\pm0.084}$ | $\underline{80.40}_{\pm26.80}$ | $\mathbf{25.76}_{\pm2.33}$ | $\underline{0.797}_{\pm0.045}$ | $\underline{0.255}_{\pm0.095}$ | $134.40_{\pm15.00}$ |
| | DCDP | $28.52_{\pm2.50}$ | $0.892_{\pm0.19}$ | $0.167_{\pm0.92}$ | $108.00_{\pm27.00}$ | $24.25_{\pm2.25}$ | $0.778_{\pm0.14}$ | $0.287_{\pm0.089}$ | $\underline{102.40}_{\pm31.20}$ |
| | SITCOM (ours) | $\underline{30.67}_{\pm3.10}$ | $\mathbf{0.915}_{\pm0.064}$ | $\underline{0.122}_{\pm0.102}$ | $\mathbf{28.50}_{\pm5.40}$ | $\underline{25.45}_{\pm2.78}$ | $\mathbf{0.808}_{\pm0.065}$ | $\mathbf{0.246}_{\pm0.088}$ | $\mathbf{84.00}_{\pm24.00}$ |
| NDB | DPS | $23.42_{\pm2.15}$ | $0.757_{\pm0.042}$ | $0.279_{\pm0.067}$ | $93.00_{\pm26.40}$ | $22.57_{\pm0.67}$ | $0.778_{\pm0.067}$ | $0.310_{\pm0.102}$ | $141.00_{\pm27.00}$ |
| | DAPS | $28.23_{\pm1.55}$ | $0.833_{\pm0.052}$ | $0.155_{\pm0.041}$ | $\underline{85.20}_{\pm24.60}$ | $27.65_{\pm1.2}$ | $0.822_{\pm0.056}$ | $0.169_{\pm0.044}$ | $128.40_{\pm25.20}$ |
| | DCDP | $28.78_{\pm1.44}$ | $0.827_{\pm0.08}$ | $0.162_{\pm0.04}$ | $92.00_{\pm27.00}$ | $26.56_{\pm1.09}$ | $0.803_{\pm0.06}$ | $0.182_{\pm0.05}$ | $\underline{89.00}_{\pm21.60}$ |
| | DMPlug | $\mathbf{30.31}_{\pm1.24}$ | $\underline{0.901}_{\pm0.051}$ | $\mathbf{0.142}_{\pm0.062}$ | $182.4_{\pm32.00}$ | – | – | – | – |
| | RED-diff | – | – | – | – | $\mathbf{29.51}_{\pm0.76}$ | $\underline{0.828}_{\pm0.08}$ | $\underline{0.211}_{\pm0.05}$ | $\mathbf{31.15}_{\pm12.40}$ |
| | SITCOM (ours) | $\underline{30.12}_{\pm0.68}$ | $\mathbf{0.903}_{\pm0.042}$ | $\underline{0.145}_{\pm0.037}$ | $33.45_{\pm9.40}$ | $\underline{28.78}_{\pm0.79}$ | $\mathbf{0.832}_{\pm0.056}$ | $\mathbf{0.16}_{\pm0.048}$ | $75.00_{\pm27.00}$ |
| HDR | DPS | $22.88_{\pm1.25}$ | $0.722_{\pm0.056}$ | $0.264_{\pm0.089}$ | $87.00_{\pm20.40}$ | $19.33_{\pm1.45}$ | $0.688_{\pm0.067}$ | $0.503_{\pm0.132}$ | $145.20_{\pm27.60}$ |
| | RED-diff | – | – | – | – | $23.45_{\pm0.54}$ | $0.746_{\pm0.052}$ | $0.257_{\pm0.045}$ | $\mathbf{24.4}_{\pm5.00}$ |
| | DAPS | $\underline{27.12}_{\pm0.89}$ | $\underline{0.825}_{\pm0.056}$ | $\underline{0.166}_{\pm0.078}$ | $75.00_{\pm21.00}$ | $26.30_{\pm1.02}$ | $\underline{0.792}_{\pm0.046}$ | $\underline{0.177}_{\pm0.089}$ | $130.80_{\pm33.00}$ |
| | SITCOM (ours) | $\mathbf{27.98}_{\pm1.06}$ | $\mathbf{0.832}_{\pm0.052}$ | $\mathbf{0.158}_{\pm0.032}$ | $31.20_{\pm8.20}$ | $\mathbf{26.97}_{\pm0.87}$ | $\mathbf{0.821}_{\pm0.045}$ | $\mathbf{0.167}_{\pm0.052}$ | $\underline{92.40}_{\pm21.00}$ |

Table 1: Average PSNR, SSIM, LPIPS, and run-time (seconds) with $\sigma_{\mathbf{y}} = 0.05$. The best (resp. second-best) results are bolded (resp. underlined). Values after ± represent the standard deviation. Dashes indicate cases where a method did not consider a dataset.

# 4. Experimental Results

## 4.1. Settings

Our setup for the image restoration problems and noise levels largely follows DPS (Chung et al., 2023b). For linear IPs, we evaluate super resolution (SR), Gaussian deblurring (GDB), motion deblurring (MDB), box inpainting (BIP), and random inpainting (RIP). For non-linear IPs, we evaluate phase retrieval (PR), non-uniform deblurring (NDB), and high dynamic range (HDR). For phase retrieval, the run-time is reported for the best result out of four independent runs (applied to SITCOM and baselines), consistent with (Chung et al., 2023b; Zhang et al., 2025). See Appendix G for a discussion on phase retrieval, and how SITCOM-ODE

(Appendix F) is empirically more stable than SITCOM and all other baselines on PR. For baselines, we use DPS (Chung et al., 2023b), DDNM (Wang et al., 2022), RED-diff (Mardani et al., 2024), PGDM (Song et al., 2023b), DCDP (Li et al., 2024), DMPlug (Wang et al., 2024), and DAPS (Zhang et al., 2025). For latent space DMs, in Appendix E, we introduce Latent-SITCOM and compare the performance to ReSample (Song et al., 2023a) and Latent-DAPS (Zhang et al., 2025). We use 100 test images from the validation set of FFHQ (Karras et al., 2019) and 100 test images from the validation set of ImageNet (Deng et al., 2009) for which the FFHQ-trained and ImageNet-trained DMs are given in (Chung et al., 2023b) and (Dhariwal & Nichol, 2021), respectively, following the previous convention.

For evaluation metrics, we use PSNR, SSIM (Wang et al., 2004), and LPIPS (Zhang et al., 2018). All experiments were conducted using a **single RTX5000 GPU** machine. In Appendix C, we show the impact of each individual consistency on the performance of SITCOM. Ablation studies are given in Appendix J. For SITCOM, Table 17 in Appendix K.1 lists all the hyper-parameters used for every task. Our code is available online.[3]

### 4.2. Main Results

In Table 1, we present quantitative results and run-time (seconds). Columns 3 to 6 (resp. 7 to 10) correspond to the FFHQ (resp. ImageNet) dataset. The table covers 8 tasks, 3 restoration quality metrics, and 2 datasets, totaling 51 results. Among these, SITCOM reports the best performance in 42 out of 51 cases. In the remaining 8, SITCOM achieved the second-best performance.

Generally, SITCOM demonstrates strong reconstruction capabilities across most tasks. SITCOM reports a PSNR improvement of over 1 dB when compared to the second best for the tasks of RIP with FFHQ (with nearly 1/3 of the run-time of DAPS), and in the task of NDB with ImageNet (with nearly one minute less run-time than DAPS and excluding RED-diff).

There are tasks where in terms of PSNR, we present a slight improvement (less than 1 dB) when compared to the second best scheme. However, we achieve a great speed-up. Examples include (*i*) SR where SITCOM requires nearly half of the run-time when compared to DMPlug for FFHQ and PGDM for ImageNet, and (*ii*) HDR where, when compared to DAPS, SITCOM requires less than half the time for FFHQ, and approximately 40 seconds less time for ImageNet. This observation is noted in nearly all other tasks.

There are 9 cases where we report the second best results. In these cases, we slightly under-perform in one or two of the three restoration quality metrics (i.e., PNSR, SSIM, and LPIPS). An example of this case is GDB on ImageNet where we report the best LPIPS but the second best PSNR (under-performing by 0.66 dB) and SSIM (under-performing by 0.025).

For PR, DAPS's PSNR is higher than SITCOM's by a small margin. However, in Appendix F, we show that SITCOM with ODE solver achieves better PSNRs than DAPS for the task of PR at a cost of increased run time (see Table 6).

In terms of run-time, SITCOM outperforms DPS, DAPS, DDNM, and DMPlug in almost all cases, as shown in Table 1. However, DCDP, REDdiff, and PGDM often exhibit faster run times compared to SITCOM. This discrepancy arises because SITCOM's default hyperparameter set-

tings prioritize achieving the highest possible PSNR, which comes at the cost of increased computational time. For instance, SITCOM demonstrates significant improvements of over 3dB in SR, RIP, and GDB for FFHQ, as well as in RIP, GDB, and HDR for ImageNet, when compared to DCDP, REDdiff, and PGDM. To ensure a fairer comparison, we provide additional results in Appendix I.2, where all methods are constrained to achieve the same PSNR. Under these conditions, SITCOM once again proves to be faster than DCDP, REDdiff, and PGDM.

SITCOM-MRI results are given in Appendix H, and visual examples are provided in Appendix L.

## 5. Conclusion

We introduced step-wise backward consistency with network regularization and formulated three conditions to achieve measurement- and diffusion-consistent trajectories for solving inverse imaging problems using pre-trained diffusion models. These conditions formed the base of our optimization-based sampling method, optimizing the diffusion model input at every sampling step for improved efficiency and measurement consistency. Experiments across eight image restoration tasks and one medical imaging reconstruction task show that our sampler consistently matches or outperforms state-of-the-art baselines, under different measurement noise levels.

## Impact Statement

The Step-wise Triple-Consistent Sampling (SITCOM) method improves diffusion models for inverse problems by introducing three consistency conditions. SITCOM accelerates sampling by reducing reverse steps while ensuring measurement, forward, and network-regularized backward diffusion consistency. This enables efficient, robust image restoration, particularly in noisy settings, and advances optimization-based approaches in generative modeling.

## Acknowledgments

This work was supported by the National Science Foundation (NSF) under grants CCF-2212065, CCF-2212066, BCS-2215155, and CAREER Award CCF-2442240. The authors would like to thank Xiang Li (University of Michigan) for discussions on compressed sensing using generative models. The authors would also like to thank Michael T. McCann (Los Alamos National Laboratory) for valuable feedback about the setting of our optimization problem.

---

[3] https://github.com/sjames40/SITCOM

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

# Appendix

In the Appendix, we start by showing the equivalence between the second formula in (2) and (5) (Appendix A). Then, we discuss the known limitations and future extensions of SITCOM (Appendix B). Subsequently, we present experiments to highlight the impact of the proposed backward consistency and other consistencies in SITCOM (Appendix C). Appendix D provides further insights to aid in understanding SITCOM. In Appendix E and Appendix F, we present the latent and ODE versions of SITCOM, respectively. This is followed by a discussion on phase retrieval (Appendix G). Appendix H shows the results of applying SITCOM on multicoil Magnetic Resonance Imaging (MRI). In Appendix I, we provide further comparison results, and in Appendix J, we perform ablation studies to examine the effects of the stopping criterion and other components/hyper-parameters in SITCOM. Appendix K covers the implementation details of tasks, baselines, and SITCOM's hyper-parameters. followed by visual examples of restored/reconstructed images (Appendix L).

## A. Derivation of Equation (5)

From (Luo, 2022), we have

$$\mathbf{s}_\theta(\mathbf{x}_t, t) = -\frac{1}{\sqrt{1 - \bar{\alpha}_t}} \boldsymbol{\epsilon}_\theta(\mathbf{x}_t, t) . \tag{13}$$

Rearranging the Tweedie's formula in (3) to solve for $\boldsymbol{\epsilon}_\theta(\mathbf{x}_t, t)$ yields

$$\boldsymbol{\epsilon}_\theta(\mathbf{x}_t, t) = \frac{\mathbf{x}_t - \sqrt{\bar{\alpha}_t} \hat{\mathbf{x}}_0(\mathbf{x}_t)}{\sqrt{1 - \bar{\alpha}_t}} . \tag{14}$$

Now, we substitute into the recursive equation for $\mathbf{x}_{t-1}$:

$$
\begin{aligned}
\mathbf{x}_{t-1} &= \frac{1}{\sqrt{1 - \beta_t}} \left[ \mathbf{x}_t + \beta_t \mathbf{s}_\theta(\mathbf{x}_t, t) \right] + \sqrt{\beta_t} \boldsymbol{\eta}_t \\
&= \frac{1}{\sqrt{1 - \beta_t}} \left[ \mathbf{x}_t + \beta_t \left( -\frac{1}{\sqrt{1 - \bar{\alpha}_t}} \boldsymbol{\epsilon}_\theta(\mathbf{x}_t, t) \right) \right] + \sqrt{\beta_t} \boldsymbol{\eta}_t \\
&= \frac{1}{\sqrt{1 - \beta_t}} \left[ \mathbf{x}_t - \frac{\beta_t}{\sqrt{1 - \bar{\alpha}_t}} \boldsymbol{\epsilon}_\theta(\mathbf{x}_t, t) \right] + \sqrt{\beta_t} \boldsymbol{\eta}_t \\
&= \frac{1}{\sqrt{1 - \beta_t}} \left[ \mathbf{x}_t - \frac{\beta_t}{\sqrt{1 - \bar{\alpha}_t}} \left( \frac{\mathbf{x}_t - \sqrt{\bar{\alpha}_t} \hat{\mathbf{x}}_0(\mathbf{x}_t)}{\sqrt{1 - \bar{\alpha}_t}} \right) \right] + \sqrt{\beta_t} \boldsymbol{\eta}_t \\
&= \frac{1}{\sqrt{1 - \beta_t}} \left[ \mathbf{x}_t - \frac{\beta_t}{1 - \bar{\alpha}_t} \left( \mathbf{x}_t - \sqrt{\bar{\alpha}_t} \hat{\mathbf{x}}_0(\mathbf{x}_t) \right) \right] + \sqrt{\beta_t} \boldsymbol{\eta}_t \\
&= \frac{1}{\sqrt{1 - \beta_t}} \left[ \left( 1 - \frac{\beta_t}{1 - \bar{\alpha}_t} \right) \mathbf{x}_t + \frac{\sqrt{\bar{\alpha}_t} \beta_t}{1 - \bar{\alpha}_t} \hat{\mathbf{x}}_0(\mathbf{x}_t) \right] + \sqrt{\beta_t} \boldsymbol{\eta}_t \\
&= \frac{(1 - \bar{\alpha}_t - \beta_t)}{\sqrt{1 - \beta_t} (1 - \bar{\alpha}_t)} \mathbf{x}_t + \frac{\sqrt{\bar{\alpha}_t} \beta_t}{\sqrt{1 - \beta_t} (1 - \bar{\alpha}_t)} \hat{\mathbf{x}}_0(\mathbf{x}_t) + \sqrt{\beta_t} \boldsymbol{\eta}_t \\
&= \frac{(\alpha_t - \bar{\alpha}_t)}{\sqrt{\alpha_t} (1 - \bar{\alpha}_t)} \mathbf{x}_t + \frac{\sqrt{\bar{\alpha}_t} \beta_t}{\sqrt{\alpha_t} (1 - \bar{\alpha}_t)} \hat{\mathbf{x}}_0(\mathbf{x}_t) + \sqrt{\beta_t} \boldsymbol{\eta}_t \\
&= \frac{\left( \sqrt{\alpha_t} - \sqrt{\alpha_t} \bar{\alpha}_{t-1} \right)}{1 - \bar{\alpha}_t} \mathbf{x}_t + \frac{\sqrt{\bar{\alpha}_{t-1}} \beta_t}{1 - \bar{\alpha}_t} \hat{\mathbf{x}}_0(\mathbf{x}_t) + \sqrt{\beta_t} \boldsymbol{\eta}_t \\
&= \frac{\sqrt{\alpha_t} (1 - \bar{\alpha}_{t-1})}{1 - \bar{\alpha}_t} \mathbf{x}_t + \frac{\sqrt{\bar{\alpha}_{t-1}} \beta_t}{1 - \bar{\alpha}_t} \hat{\mathbf{x}}_0(\mathbf{x}_t) + \sqrt{\beta_t} \boldsymbol{\eta}_t ,
\end{aligned}
$$

which is equivalent to the second formula in (2).

## B. Limitations & Future Work

The stated conditions and proposed sampler are limited to the non-blind setting, as SITCOM assumes full access to the forward model, unlike works such as (Chung et al., 2023a), which perform both image restoration and forward model estimation. Additionally, SITCOM uses 2D diffusion models for 2D image reconstruction/restoration only. Extending SITCOM to the 3D setting may require specialized regularization and modifications.

For future work, we aim to address the above two settings (3D reconstruction and/or blind settings) and explore its applicability in medical 3D image reconstruction.

## C. Impact of Each Individual Consistency on SITCOM's Performance

Here, we show the effect of removing each of the individual consistencies to demonstrate their necessity.

### C.1. Impact of the Backward Consistency

In SITCOM, a key contribution is the imposition of *step-wise* backward consistency, which allows us to fully exploit the implicit regularization provided by the network. Removing stepwise backward consistency is equivalent to removing the network regularization constraint $\hat{\mathbf{x}}_0 = f(\mathbf{v}_t; \theta, t)$ from SITCOM. More specifically, in our algorithm, the optimization variable becomes the output of the pre-trained network for which, given some $\mathbf{x}_t$, the steps at sampling time $t$ are:

$$\hat{\mathbf{x}}_0' = \text{argmin}_{\mathbf{x}'} \left\| \mathcal{A}(\mathbf{x}') - \mathbf{y} \right\|_2^2, \tag{15a}$$

$$\mathbf{x}_{t-1} = \sqrt{\bar{\alpha}_{t-1}} \hat{\mathbf{x}}_0' + \sqrt{1 - \bar{\alpha}_{t-1}} \boldsymbol{\eta}_t, \quad \boldsymbol{\eta}_t \sim \mathcal{N}(\mathbf{0}, \mathbf{I}), \tag{15b}$$

where the initialization of $\mathbf{x}'$ in (15a) is set to $f(\mathbf{x}_t; t, \boldsymbol{\epsilon}_\theta)$. We call the resulting algorithm as "SITCOM without backward consistency". In (15a), we already obtain an estimated image. Therefore, there is no need for a second step. Furthermore, as the pre-trained model here is only used for initializing the optimization in (15a), SITCOM without backward consistency does not require to back-propagate through the pre-trained DM network which will subsequently result in reduced run-times. However, the steps in (15) require more sampling iterations to converge to decent PSNRs. This statement will be supported in the next subsubsections.

**We remark that this reduced algorithm is essentially equivalent to the Resample Algorithm (Song et al., 2023a) in the *pixel* space. In other words, Resample in the pixel space resembles SITCOM without backward consistency.**

In this subsection, we will show the impact of the step-wise DM network regularization of the backward consistency by examining the results of (*i*) all sampling steps, and (*ii*) intermediate steps.

#### C.1.1. FULL SAMPLING STEPS STUDY

Here, we run SITCOM vs. SITCOM without backward consistency (i.e., Algorithm 1 vs. the steps in (15)) for the task of super resolution. We report the average PSNR and run-time (seconds) of 20 FFHQ test images with $K = 1000$ and $N \in \{20, 100\}$. We select a large $K$ to allow the optimizer to converge. Results are given in Table 2. As observed, SITCOM without backward consistency indeed requires less run-time but the PSNRs are significantly lower than SITCOM (more than 5 dB for both cases) even if run until convergence (i.e., with very large $K$).

| Method | $N = 20$ | | $N = 100$ | |
|---|---|---|---|---|
| | PSNR | Run-time | PSNR | Run-time |
| SITCOM | 31.34 | 608.2 | 31.39 | 2418.2 |
| SITCOM without Backward Consistency | 25.97 | 372.4 | 26.1 | 1405.7 |

Table 2: Impact of the backward consistency in SITCOM: Average PSNR and run-time results (in seconds) of 20 FFHQ test images at using different values of $N$ with $K = 1000$, reported by running Algorithm 1 with the optimization in (S$_1$) (first row) vs. the optimization in (15a) (second row).

#### C.1.2. INTERMEDIATE SAMPLING STEPS VISUALIZATIONS

First, for the box-painting task, we compare SITCOM with optimizing over the output of the DM network (i.e., the steps in (15)) at time steps $t' \in \{200, 400, 600\}$. More specifically, for each case (selection of $t'$), we start from $t = T$ and

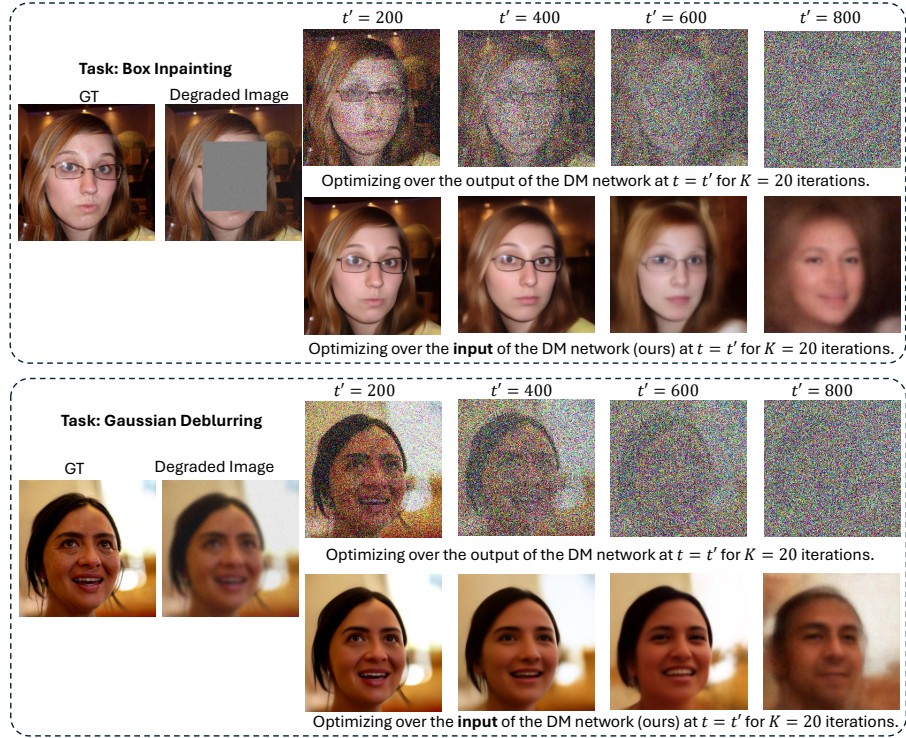

Figure 3: Results of applying optimization-based measurement consistency, for which the optimization variable is the DM output (resp. input), are shown in the first (resp. second) row for each task: Box Inpainting (*top*) and Gaussian Deblurring (*bottom*).

run SITCOM with a step size of $\lfloor \frac{T}{N} \rfloor$. At $t = t'$, given $\mathbf{x}_{t'}$, we perform two separate optimizations with intializing the optimization variable as $\mathbf{x}_{t'}$: one iteratively over the DM network input (i.e., ($S_1$)) and another iteratively over the DM network output (i.e., (15a)), both running until convergence (i.e., when the loss stops decreasing). Figure 1 shows the results at different time steps. The consistency between the ground truth and the unmasked regions of the estimated images suggest the convergence of the measurement consistency. As observed, SITCOM produces significantly less artifacts in the masked region when compared to optimizing over the output. This is evident both at earlier time steps ($t' = 600$) and later steps ($t' = 400$ and $t' = 200$).

For the second experiment, the goal is to show that SITCOM requires much smaller number of optimization steps to remove the noise as compared to SITCOM without backward consistency. The results are given in Figure 3, where we repeat the above experiment with two tasks: Box-inpainting (*top*) and Gaussian Deblurring (*bottom*), this time using a fixed number of optimization steps for both SITCOM, and SITCOM without backward consistency. Specifically, we run SITCOM from $t = T$ to $t = t' + 1$. Then, we apply $K = 20$ iterations (the setting in SITCOM) in ($S_1$), and $K = 20$ for (15a) where measurement noise is $\sigma_{\mathbf{y}} = 0.05$. As shown, compared to SITCOM without backward consistency, SITCOM significantly reduces noise across all considered $t'$, underscoring the effect of the proposed step-wise network regularization for backward diffusion consistency.

## C.2. Impact of the iterative step-wise measurement/data consistency

While all DM-IP methods aim to impose data consistency in the final solution, not all enforce *step-wise* data consistency. This step-wise consistency requires that the estimate $\hat{\mathbf{x}}_0$ computed using Tweedie's formula *at each intermediate step* remains consistent with the data. In SITCOM, step-wise measurement consistency is enforced by minimizing the data misfit term $\|\mathcal{A}(\hat{\mathbf{x}}_0(\mathbf{v}_t)) - \mathbf{y}\|_2^2$ at each step. However, if the minimizer is not found and only one gradient update of $\|\mathcal{A}(\hat{\mathbf{x}}_0(\mathbf{v}_t)) - \mathbf{y}\|_2^2$ with respect to $\mathbf{v}_t$ is performed (as in DPS), step-wise data consistency is no longer enforced, although global data consistency at $t = 0$ is still maintained. Therefore, we refer to SITCOM with $K = 1$ (i.e., performing only one gradient update of the data misfit) as SITCOM without data consistency and compare it to SITCOM itself. Note that the former is also equivalent to DPS plus resampling.

More specifically, we compare SITCOM with $K = 1$ vs. $K = 20$ (converged) for various sampling steps ($N$). The study

uses 20 FFHQ images with $\sigma_{\mathbf{y}} = 0.05$. Average PSNR and run-time (in seconds) for the tasks SR and NDB are given in Table 3.

| | Super Resolution | | | | Non-linear Deblurring | | | |
| | $K = 1$ | | $K = 20$ | | $K = 1$ | | $K = 20$ | |
| $N$ | PSNR | Run-time | PSNR | Run-time | PSNR | Run-time | PSNR | Run-time |
|---|---|---|---|---|---|---|---|---|
| 10 | 11.28 | 0.99 | 27.75 | 19.12 | 12.12 | 0.99 | 26.92 | 19.02 |
| 20 | 11.41 | 2.11 | 30.40 | 35.20 | 12.30 | 2.30 | 30.27 | 36.05 |
| 30 | 11.44 | 3.12 | 30.88 | 55.04 | 12.32 | 3.40 | 30.72 | 57.05 |
| 100 | 17.04 | 9.14 | - | - | 16.46 | 9,12 | - | - |
| 1000 | 25.44 | 60.43 | - | - | 24.89 | 60.45 | - | - |

Table 3: Impact of the step-wise iterative measurement consistency in SITCOM: Average PSNR/run-time results of 20 FFHQ test images at using different values of $N$ and $K$ to show the impact of using multiple gradient updates in applying the step-wise measurement consistency in SITCOM.

We observe that $K = 20$ (i.e., imposing the step-wise data consistency) yields better result than $K = 1$ (i.e., not imposing the step-wise data consistency). In addition, for $K = 1$ (i.e., DPS+resampling), achieving decent results requires $N = 1000$, leading to longer run-times.

### C.3. Impact of the step-wise forward consistency

This experiment examines the impact of forward consistency imposed through the resampling in step 3 of SITCOM. Specifically, we compare SITCOM versus SITCOM with replacing the resampling formula with the traditional sampling formula in (5) at each sampling step.

We use 20 FFHQ test images with $\sigma_{\mathbf{y}} = 0.05$, setting $N = K = 20$. Table 4 shows average PSNR/runtime (in seconds) for one linear and one non-linear task. As observed, the use of resampling formula is more effective than the standard sampling,

| Method | Mapping to $t-1$ | Super Resolution | | Non-linear Deblurring | |
| | | PSNR | Run-time | PSNR | Run-time |
|---|---|---|---|---|---|
| SITCOM | Equation (S$_3$) | 30.40 | 35.20 | 30.27 | 36.05 |
| SITCOM without Forward Consistency (resampling) | Equation (6) | 20.78 | 35.12 | 22.42 | 36.22 |

Table 4: Impact of the forward consistency in SITCOM: Average PSNR and run-time (seconds) results of 20 FFHQ test images using two re-mappings: First is (S$_3$) (first row), and second is (6).

demonstrating the effect of the forward consistency in SITCOM.

## D. Further Insights & Relation to Existing Studies

Here, we discuss similarities and differences between DPS and SITCOM which leads to providing further insights into why our proposed sampler can (*i*) achieve competitive quantitative results, and (*ii*) allow for arbitrary sampling iterations.

DPS (Algorithm 2) is known to approximately converge to the posterior distribution $p(\mathbf{x}|\mathbf{y})$ at $t = 0$.

**Algorithm 2: DPS - Gaussian (Chung et al., 2023b)**

**Require:** $N, \mathbf{y}, \{\zeta_i\}_{i=1}^N, \{\tilde{\sigma}_i\}_{i=1}^N$
1: $\mathbf{x}_N \sim \mathcal{N}(0, I)$
2: **for** $i = N - 1$ to $0$ **do**
3:     $\hat{\boldsymbol{s}} \leftarrow \boldsymbol{s}_\theta(\mathbf{x}_i, i)$
4:     $\hat{\mathbf{x}}_0 \leftarrow \frac{1}{\sqrt{\bar{\alpha}_i}}(\mathbf{x}_i + (1 - \bar{\alpha}_i)\hat{\boldsymbol{s}})$
5:     $\mathbf{z} \sim \mathcal{N}(0, I)$
6:     $\mathbf{x}'_{i-1} \leftarrow \frac{\sqrt{\alpha_i}(1-\bar{\alpha}_{i-1})}{1-\bar{\alpha}_i}\mathbf{x}_i + \frac{\sqrt{\bar{\alpha}_{i-1}\beta_i}}{1-\bar{\alpha}_i}\hat{\mathbf{x}}_0 + \tilde{\sigma}_i\mathbf{z}$
7:     $\mathbf{x}_{i-1} \leftarrow \mathbf{x}'_{i-1} - \zeta_i\nabla_{\mathbf{x}_i}\|\mathbf{y} - \mathcal{A}(\hat{\mathbf{x}}_0)\|_2^2$
8: **end for**
9: **return** $\hat{\mathbf{x}}$

**Algorithm 3: DPS with resampling**

**Require:** $N, \mathbf{y}, \{\zeta_i\}_{i=1}^N, \{\tilde{\sigma}_i\}_{i=1}^N$
1: $\mathsf{x}_N \sim \mathcal{N}(0, I)$
2: **for** $i = N - 1$ to $0$ **do**
3:     $\hat{\boldsymbol{s}} \leftarrow \boldsymbol{s}_\theta(\mathbf{x}_i, i)$
4:     $\hat{\mathbf{x}}_0 \leftarrow \frac{1}{\sqrt{\bar{\alpha}_i}}(\mathbf{x}_i + (1 - \bar{\alpha}_i)\hat{\boldsymbol{s}})$
5:     $\mathbf{z} \sim \mathcal{N}(0, I)$
6:     $\mathbf{v}_i \leftarrow \mathbf{x}_i - \zeta_i\nabla_{\mathbf{x}_i}\|\mathbf{y} - \mathcal{A}(\hat{\mathbf{x}}_0)\|_2^2$
7:     $\mathbf{x}'_{i-1} \leftarrow \sqrt{\bar{\alpha}_{i-1}}\hat{\mathbf{x}}_0(\mathbf{v}_i) + \sqrt{1 - \bar{\alpha}_{i-1}}\mathbf{z}$
8: **end for**
9: **return** $\hat{\mathbf{x}}$

**Proposition D.1.** *SITCOM with $K = 1$ is DPS with the resampling formula in* (7).

*Proof.* First, we replace the $\mathbf{x}_i$ in line 6 of DPS (Algorithm 2) with its approximation $\mathbf{x}_i \approx \sqrt{\bar{\alpha}_i}\hat{\mathbf{x}}_0 + \sqrt{1 - \bar{\alpha}_i}\mathbf{z}$, where $\mathbf{z} \sim \mathcal{N}(0, \mathbf{I})$. Under this substitution, line 6 becomes $\mathbf{x}'_{i-1} = \sqrt{\bar{\alpha}_{i-1}}\hat{\mathbf{x}}_0 + \sqrt{1 - \bar{\alpha}_{i-1}}\mathbf{z}$, which corresponds to the resampling formula. The resulting algorithm is referred to as **DPS with resampling** (DPS+resampling).

To better illustrate its connection to SITCOM, we rename the variable $\mathbf{x}'_{i-1}$ to $\mathbf{v}_i$ and swap the resampling formula with line 7 of DPS. These changes do not alter the algorithm's outcome. The resulting is an equivalent form of DPS+resampling and is presented in **Algorithm 3**.

We observe that line 6 in the Algorithm 3 corresponds to the gradient descent step for the objective function $\|\mathbf{y} - \mathcal{A}(\hat{\mathbf{x}}_0(\mathbf{x}_i))\|_2^2$ with respect to $\mathbf{x}_i$, and line 7 is the resampling. Compared to Algorithm 1, we see that DPS+resampling is equivalent to SITCOM with $K = 1$. $\qquad\square$

**Remark 3.** From Proposition 1, we observe that each iteration of DPS performs a single gradient update on the objective function $\|\mathbf{y} - \mathcal{A}(\hat{\mathbf{x}}_0(\mathbf{x}_i))\|_2^2$. By grouping $K$ consecutive sampling steps together, the process effectively performs $K$ gradient updates. These updates correspond to very similar objective functions due to the continuity of the diffusion model, where $\boldsymbol{s}_\theta(\mathbf{x}, t) \approx \boldsymbol{s}_\theta(\mathbf{x}, t')$ for $t \approx t'$.

Specifically, within each group of $K$ consecutive sampling steps, the associated time steps are closely spaced, say they are all within an interval $(t_1, t_2)$ with $t_2 - t_1 \ll 1$. Due to the continuity of the diffusion model, the $\boldsymbol{s}_\theta(\mathbf{x}, t)$ term in $\hat{\mathbf{x}}_0$ can be approximated by its value at the fixed time $t_2$, i.e., $\boldsymbol{s}_\theta(\mathbf{x}, t) \approx \boldsymbol{s}_\theta(x, t_2)$ for all $t \in (t_1, t_2)$. With this approximation, the objective function $\|\mathbf{y} - \mathcal{A}(\hat{\mathbf{x}}_0(\mathbf{x}_i))\|_2^2$ depends solely on $\mathbf{x}_i$, rather than both $\mathbf{x}_i$ and $t$.

This simplification implies that the $K$ gradient updates now act on the same objective function. As a result, optimization can be accelerated using Adam. The use of Adam is a key reason why SITCOM achieves faster convergence compared to DPS, as the latter relies solely on gradient descent. Finally, as shown in ReSample (Song et al., 2023a), resampling can be performed rather sparsely. In SITCOM, resampling is performed only once every K iterations, further enhancing its efficiency.

It is important to note that this explanation provides a high-level intuition to aid in understanding SITCOM, rather than a rigorous proof of its performance.

# E. Latent SITCOM

In this section, we extend SITCOM to latent diffusion models. Define $\mathcal{E} : \mathbb{R}^n \to \mathbb{R}^r$ (parameterized by $\phi$) and $\mathcal{D} : \mathbb{R}^r \to \mathbb{R}^n$ (parameterized by $\psi$) as the pre-trained encoder and decoder, respectively, with $r \ll n$. Let $\epsilon : \mathbb{R}^r \times \{0, \ldots, T\} \to \mathbb{R}^r$ (parameterized by $\theta'$) be a pre-trained DM in the latent space. We note that $\mathcal{E}$ is only needed during training. At inference, given some $\mathbf{z}_t \in \mathbb{R}^r$, the steps in latent SITCOM are formulated as

$$\hat{\mathbf{w}}_t := \underset{\mathbf{w}'_t}{\operatorname{argmin}} \left\| \mathcal{A}\Big(\mathcal{D}_\psi\big(\frac{1}{\sqrt{\bar{\alpha}_t}}\big[\mathbf{w}'_t - \sqrt{1 - \bar{\alpha}_t}\epsilon_{\theta'}(\mathbf{w}'_t, t)\big]\big)\Big) - \mathbf{y} \right\|_2^2 + \lambda \left\| \mathbf{w}'_t - \mathbf{z}_t \right\|_2^2, \tag{SL_1}$$

$$\hat{\mathbf{z}}_0' := \frac{1}{\sqrt{\bar{\alpha}_t}}\left[\hat{\mathbf{w}}_t - \sqrt{1 - \bar{\alpha}_t}\epsilon_{\theta'}(\hat{\mathbf{w}}_t, t)\right], \tag{SL$_2$}$$

$$\mathbf{z}_{t-1} = \sqrt{\bar{\alpha}_{t-1}}\hat{\mathbf{z}}_0' + \sqrt{1 - \bar{\alpha}_{t-1}}\zeta_t, \ \ \zeta_t \sim \mathcal{N}(\mathbf{0}, \mathbf{I}_r). \tag{SL$_3$}$$

| Task | Method | FFHQ | | | ImageNet | | |
|------|--------|------|------|------|------|------|------|
| | | PSNR ($\uparrow$) | LPIPS ($\downarrow$) | run-time ($\downarrow$) | PSNR ($\uparrow$) | LPIPS ($\downarrow$) | run-time ($\downarrow$) |
| Super Resolution 4× | Latent-DAPS (Zhang et al., 2025) | 27.48 | 0.192 | 84.24 | 25.06 | 0.343 | 125.40 |
| | ReSample (Song et al., 2023a) | 23.33 | 0.392 | 194.30 | 22.19 | 0.370 | 265.30 |
| | Latent-SITCOM (ours) | **27.98** | **0.142** | **65.00** | **26.35** | **0.232** | **123.20** |
| Box In-Painting | Latent-DAPS (Zhang et al., 2025) | 23.99 | 0.194 | 83.00 | 17.19 | 0.340 | 115.20 |
| | ReSample (Song et al., 2023a) | 20.06 | 0.184 | 189.45 | 18.29 | **0.262** | 226.24 |
| | Latent-SITCOM (ours) | **24.22** | **0.177** | **58.20** | **20.88** | 0.314 | **114.00** |
| Random In-Painting | Latent-DAPS (Zhang et al., 2025) | 30.71 | 0.146 | 93.65 | 27.89 | 0.202 | 128 |
| | Latent-SITCOM (ours) | **31.05** | **0.135** | **63.24** | **28.44** | **0.197** | **118.30** |
| Gaussian Deblurring | Latent-DAPS (Zhang et al., 2025) | 28.03 | 0.232 | 88.420 | 25.34 | 0.341 | 135.24 |
| | ReSample (Song et al., 2023a) | 26.42 | 0.255 | 186.40 | **25.97** | **0.254** | 234.10 |
| | Latent-SITCOM (ours) | **28.21** | **0.223** | **78.30** | 25.72 | 0.316 | **132.24** |
| Motion Deblurring | Latent-DAPS (Zhang et al., 2025) | 27.32 | 0.278 | 118.2 | 26.85 | 0.349 | 139.2 |
| | ReSample (Song et al., 2023a) | 27.41 | 0.198 | 236.2 | 26.94 | 0.227 | 264.2 |
| | Latent-SITCOM (ours) | **27.74** | **0.182** | **64.25** | **27.25** | **0.202** | **121.4** |
| Phase Retrieval | Latent-DAPS (Zhang et al., 2025) | **29.16** | **0.199** | **80.92** | **20.54** | 0.361 | **105.65** |
| | ReSample (Song et al., 2023a) | 21.60 | 0.406 | 204.5 | 19.24 | 0.403 | 234.25 |
| | Latent-SITCOM (ours) | 28.14 | 0.241 | 114.52 | 20.45 | **0.326** | 156.2 |
| Non-Uniform Deblurring | Latent-DAPS (Zhang et al., 2025) | 28.15 | 0.211 | 89.24 | 25.34 | 0.314 | 130.2 |
| | ReSample (Song et al., 2023a) | 28.24 | 0.185 | 172.4 | 26.20 | 0.206 | 215.2 |
| | Latent-SITCOM (ours) | **28.41** | **0.171** | **112.4** | **26.48** | **0.201** | 134.5 |
| High Dynamic Range | Latent-DAPS (Zhang et al., 2025) | **26.14** | 0.221 | **78.2** | 23.78 | 0.261 | **132.2** |
| | ReSample (Song et al., 2023a) | 25.55 | 0.182 | 178.2 | 25.11 | 0.196 | 215.2 |
| | Latent-SITCOM (ours) | 25.85 | **0.174** | 102.45 | **25.67** | **0.182** | 145.4 |

Table 5: Average PSNR, LPIPS, and run-time (seconds) results of our method and other latent-space baselines over 100 FFHQ and 100 ImageNet test images. The measurement noise setting is $\sigma_\mathbf{y} = 0.05$.

Table 5 presents the results of Latent-SITCOM for the 8 image restoration tasks as compared to ReSample (Song et al., 2023a) and Latent-DAPS (Zhang et al., 2025). All methods used the pre-trained latent model from ReSample[4].

For SITCOM, we used $N = 20$ and $\lambda = 0$ and set $K = 30$ (resp. $K = 50$) for linear (resp. non-linear) problems. Default hyper-parameter settings were used for ReSample and Latent-DAPS for each task as recommended in the authors' papers and code base.

As observed, Latent-SITCOM achieves better results than ReSample on all tasks and on both metrics (PSNR and LPIPS). When compared to Latent-DAPS, SITCOM achieves improved results on all tasks other than phase retrieval where we under-perform by 1dB on FFHQ and nearly 0.1dB on ImageNet.

In terms of run-time, Latent-SITCOM generally requires less time when compared to ReSample. As compared to Latent-DAPS, our run-time is generally less. However, there are cases where Latet-DAPS is faster than our method. An example of this is MDB on FFHQ.

In summary, in this section, we show that SITCOM can be extended to the latent space, and our results are either better or very competitive when compared to two recent state-of-the-art latent-space baselines.

## F. SITCOM with ODEs for Step (2) in (S$_2$)

The Tweedie's network denoiser $f(x_t; t, \epsilon_\theta)$ in (S$_2$) can be replaced by other denoisers without affecting the main idea of imposing the three consistencies in SITCOM. In this section, we conduct a study for which we replace it with an ODE solver at each sampling step $t$. Specifically, given $\hat{\mathbf{v}}_t$ from (S$_1$), we obtain an estimated image $\hat{\mathbf{x}}_0'$ by solving following ODE

---

[4] https://github.com/soominkwon/resample

with initial values $\hat{\mathbf{v}}_t$ and $t$.

$$\frac{d\mathbf{x}_t}{dt} = -\dot{\sigma}_t \sigma_t \nabla_{\mathbf{x}_t} \log p(\mathbf{x}_t; \sigma_t) . \tag{16}$$

In (16), $\nabla_{\mathbf{x}_t} \log p(\mathbf{x}_t; \sigma_t) \approx \boldsymbol{s}_\theta(\hat{\mathbf{v}}_t, t)$ where we implement the Euler solver in (Karras et al., 2022) and select $\sigma_t = t$. This reduces (16) to solving the ODE $d\mathbf{x}_t = -t\boldsymbol{s}_\theta(\hat{\mathbf{v}}_t, t)dt$ using $N_{\text{ODE}}$ discrete steps.

| Task | Method | FFHQ | | | ImageNet | | |
|---|---|---|---|---|---|---|---|
| | | PSNR (↑) | LPIPS (↓) | run-time (↓) | PSNR (↑) | LPIPS (↓) | run-time (↓) |
| Super Resolution 4× | SITCOM | 30.68 | 0.142 | **27.00** | 26.35 | 0.232 | **67.20** |
| | SITCOM-ODE | **30.86** | **0.137** | 76.10 | **26.51** | **0.228** | 132.00 |
| Box In-Painting | SITCOM | 24.68 | 0.121 | **21.00** | 21.88 | 0.214 | **67.20** |
| | SITCOM-ODE | **24.79** | **0.118** | 81.30 | **22.44** | **0.208** | 152.00 |
| Random In-Painting | SITCOM | 32.05 | 0.095 | **27.00** | 29.60 | 0.127 | **68.40** |
| | SITCOM-ODE | **32.18** | **0.091** | 93.50 | **30.11** | **0.114** | 128.30 |
| Gaussian Deblurring | SITCOM | 30.25 | 0.135 | **27.60** | 27.40 | 0.236 | **66.00** |
| | SITCOM-ODE | **30.42** | **0.132** | 85.10 | **27.87** | **0.232** | 134.10 |
| Motion Deblurring | SITCOM | 30.34 | 0.148 | **30.00** | 28.65 | 0.189 | **88.80** |
| | SITCOM-ODE | **30.54** | **0.145** | 112.34 | **28.81** | **0.184** | 139.00 |
| Phase Retrieval | SITCOM | 30.67 | 0.122 | **28.50** | 25.45 | 0.246 | **84.00** |
| | SITCOM-ODE | **31.10** | **0.109** | 80.40 | **25.96** | **0.242** | 138.50 |
| Non-Uniform Deblurring | SITCOM | 30.12 | 0.145 | **33.45** | 28.78 | 0.16 | **75.00** |
| | SITCOM-ODE | **30.31** | **0.141** | 85.25 | **28.86** | **0.152** | 129.20 |
| High Dynamic Range | SITCOM | 27.98 | 0.158 | **31.20** | 26.97 | 0.177 | **92.40** |
| | SITCOM-ODE | **28.12** | **0.155** | 75.20 | **27.14** | **0.164** | 132.00 |

Table 6: Average PSNR, LPIPS, and run-time results of SITCOM with Tweedie's formula (i.e., Algorithm 1) vs. SITCOM with the ODE solver for (16) instead of Tweedie's in (S$_2$). Results are averaged over 100 FFHQ and 100 ImageNet test images with measurement noise level of $\sigma_{\mathbf{y}} = 0.05$. For SITCOM ODE, similar to DAPS (Zhang et al., 2025), we use $N_{\text{ODE}} = 5$.

Table 6 presents a comparison study between SITCOM with Tweedie's formula vs. SITCOM with ODE in terms of PSNR, LPIPS, and run-time.

In general, we observe that the run-time for SITCOM-ODE is significantly longer than the required run-time for SITCOM. Specifically, the run-time is either doubled (or, in some cases, tripled) for a PSNR gain of less than 1 dB. These results show the trade-off between PSNR and run-time when ODEs are used in SITCOM, as discussed in Remark 1.

It is important to note that SITCOM-ODE demonstrates greater stability for the task of PR. In particular, we observe fewer failures when selecting the best result from 4 runs for each image. See the next section for further details.

## G. Discussion on Phase Retrieval

As discussed in our experimental results section, for the phase retrieval task, we report the best results from 4 independent runs, following the convention in (Chung et al., 2023b; Zhang et al., 2025). For the phase retrieval results of Table 1, we use this approach across all baselines where the run-time is reported for one run.

The forward model for phase retrieval is adopted from DPS where the inverse problem is generally more challenging compared to other image restoration tasks. This increased difficulty arises from the presence of multiple modes that can yield the same measurements (Zhang et al., 2025).

In Figure 4, we present two examples. For each ground truth image, we show four results from which the best one was selected. In the first column, SITCOM avoids significant artifacts, while DAPS and SITCOM-ODE produce one image rotated by 180 degrees. In the second column, both SITCOM and DAPS exhibit one run with severe artifacts whereas SITCOM-ODE shows no image with extreme artifacts. The last image from SITCOM does exhibit more artifacts compared to the second worst-case result from DAPS. Additionally, the DPS results show severe perceptual differences in both cases, with artifacts being particularly noticeable in the second column.

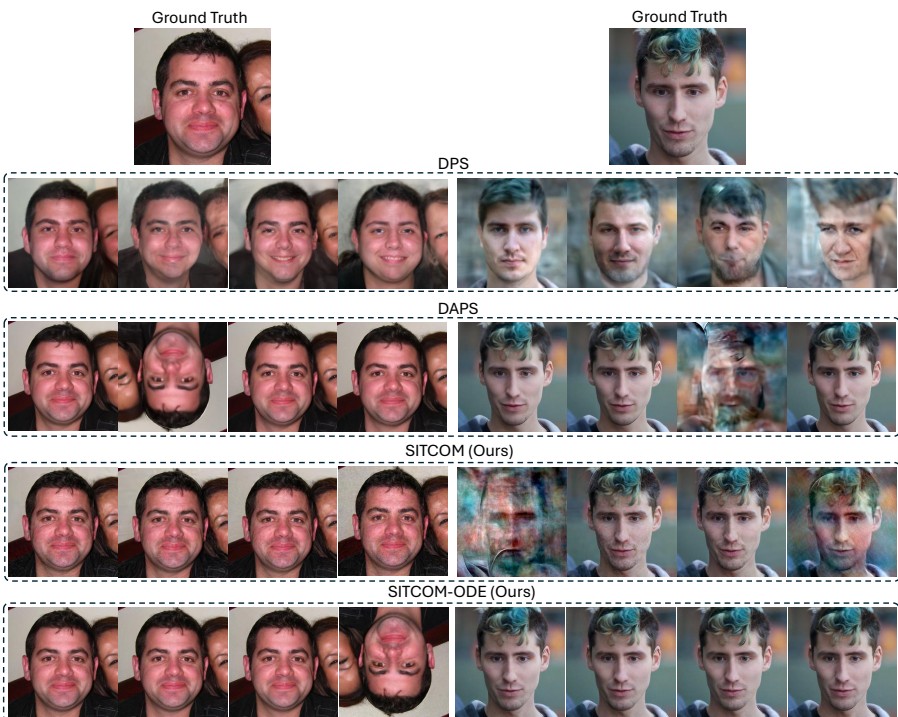

Figure 4: Results of Phase Retrieval on two images (top row) from the FFHQ dataset. Rows 2, 3, 4, and 5 correspond to the results of DPS, DAPS, and SITCOM (ours), and SITCOM-ODE (ours), respectively.

## H. SITCOM MRI Setup and Results

In this section, we extend SITCOM to 2D medical image reconstruction. In particular, we consider the linear task of multi-coil Magnetic Resonance Imaging (MRI) where the forward operator is $\mathcal{A} = \mathbf{MFS}$. Here, $\mathbf{M}$ denotes the coil-wise undersampling, $\mathbf{F}$ is the coil-by-coil Fourier transform, and $\mathbf{S}$ represents the sensitivity encoding with multiple coils, which are incorporated into the operator $\mathbf{A}$ for all scenarios, are obtained using the BART toolbox (Tamir et al., 2016). We consider two mask patterns and two acceleration factors (Ax).

For baselines, we compare against DM-based solvers (DDS (Chung et al., 2024) and Score-MRI (Chung & Ye, 2022)), supervised learning solvers (Supervised U-Net (Ronneberger et al., 2015) and E2E Varnet (Sriram et al., 2020)), and dataless approaches (Self-Guided DIP (Liang et al., 2025), Auto-encoded Sequential DIP (aSeqDIP) (Alkhouri et al., 2024a), and TV-ADMM (Uecker et al., 2014)). We use the fastMRI dataset (et al, 2019). We note that for training the Supervised U-Net (Ronneberger et al., 2015) and E2E Varnet (Sriram et al., 2020), we selected 8,000 training images from the 973 available volumes, omitting the first and last five slices from each volume. For testing, we used 50 images taken from the validation dataset.

SITCOM and the other two DM-based solvers (DDS and Score-MRI) use the pre-trained model in DDS[5]. We used the recommended parameters for baselines as given in their respective papers. For SITCOM, we used $N = 50$, $K = 10$, and $\lambda = 0$.

PSNR and SSIM results are presented in Table 7, while run-time results are provided in Table 8. As observed, SITCOM achieves the highest PSNR and SSIM values compared to the considered baselines. In terms of run-time, SITCOM requires significantly less time compared to DPS and Score-MRI. When compared to DDS (the best DM-based baseline), SITCOM achieves more than a 1 dB improvement in 3 out of the 4 considered mask patterns/acceleration factors, though it requires nearly twice the run-time. DDS is faster than SITCOM because, in DDS, back-propagation through the pre-trained network is not required, as it is in SITCOM. Visualizations are given in Figure 5.

---

[5] https://github.com/HJ-harry/DDS

| Mask Pattern | Ax. | TV-ADMM | Supervised | E2E-VarNet | Self-G DIP | aSeqDIP | Score-MRI | DPS | DDS | SITCOM |
|---|---|---|---|---|---|---|---|---|---|---|
| Uniform 1D | $\times 4$ | $27.41_{\pm 0.43}$ | $31.67_{\pm 0.89}$ | $33.06_{\pm 0.59}$ | $32.59_{\pm 0.12}$ | $32.75_{\pm 0.452}$ | $33.12_{\pm 1.18}$ | $30.56_{\pm 0.66}$ | $\underline{34.95}_{\pm 0.74}$ | $\mathbf{36.33}_{\pm 0.37}$ |
| | | $0.667_{\pm 0.17}$ | $0.847_{\pm 0.05}$ | $0.854_{\pm 0.12}$ | $0.851_{\pm 0.45}$ | $0.852_{\pm 0.08}$ | $0.849_{\pm 0.08}$ | $0.841_{\pm 0.11}$ | $\underline{0.954}_{\pm 0.10}$ | $\mathbf{0.962}_{\pm 0.08}$ |
| | $\times 8$ | $25.12_{\pm 2.11}$ | $29.24_{\pm 0.37}$ | $32.03_{\pm 0.35}$ | $32.19_{\pm 0.45}$ | $\underline{32.33}_{\pm 0.31}$ | $32.10_{\pm 2.30}$ | $30.33_{\pm 0.23}$ | $31.72_{\pm 1.88}$ | $\mathbf{32.78}_{\pm 0.64}$ |
| | | $0.535_{\pm 0.05}$ | $0.784_{\pm 0.05}$ | $0.829_{\pm 0.08}$ | $0.827_{\pm 0.16}$ | $0.831_{\pm 0.14}$ | $0.821_{\pm 0.13}$ | $0.812_{\pm 0.18}$ | $\underline{0.876}_{\pm 0.06}$ | $\mathbf{0.892}_{\pm 0.02}$ |
| Gaussian 1D | $\times 4$ | $30.55_{\pm 1.77}$ | $32.78_{\pm 0.66}$ | $34.11_{\pm 0.14}$ | $33.98_{\pm 1.25}$ | $34.28_{\pm 0.95}$ | $34.21_{\pm 1.54}$ | $32.37_{\pm 1.09}$ | $\underline{35.24}_{\pm 1.01}$ | $\mathbf{36.42}_{\pm 0.85}$ |
| | | $0.785_{\pm 0.06}$ | $0.861_{\pm 0.12}$ | $0.913_{\pm 0.12}$ | $0.904_{\pm 0.15}$ | $0.908_{\pm 0.14}$ | $0.891_{\pm 0.11}$ | $0.832_{\pm 0.15}$ | $\underline{0.963}_{\pm 0.15}$ | $\mathbf{0.969}_{\pm 0.13}$ |
| | $\times 8$ | $28.08_{\pm 1.28}$ | $31.52_{\pm 1.09}$ | $33.21_{\pm 0.29}$ | $32.81_{\pm 1.42}$ | $32.95_{\pm 0.88}$ | $32.34_{\pm 0.55}$ | $30.52_{\pm 2.32}$ | $\underline{33.32}_{\pm 1.66}$ | $\mathbf{33.99}_{\pm 1.29}$ |
| | | $0.747_{\pm 0.21}$ | $0.841_{\pm 0.12}$ | $0.868_{\pm 0.18}$ | $0.873_{\pm 0.13}$ | $0.877_{\pm 0.17}$ | $0.853_{\pm 0.11}$ | $0.833_{\pm 0.16}$ | $\underline{0.933}_{\pm 0.07}$ | $\mathbf{0.943}_{\pm 0.09}$ |

Table 7: Average PSNR and SSIM results for SITCOM and various methods for MRI reconstruction using Uniform 1D and Gaussian 1D masks with 4x and 8x acceleration factors. The measurement noise level is $\sigma_{\mathbf{y}} = 0.01$. Values past $\pm$ represent the standard deviation. Best results are bolded whereas the second-best results are underlined.

| Method | Score-MRI | DPS | DDS | SITCOM |
|---|---|---|---|---|
| Avg. Run-time (seconds) | $342.20_{\pm 41}$ | $145.52_{\pm 27}$ | $25.40_{\pm 6.1}$ | $52.30_{\pm 22}$ |

Table 8: Average run-time in seconds for SITCOM and DM-based baselines for the case of Uniform 1D and $\times 4$.

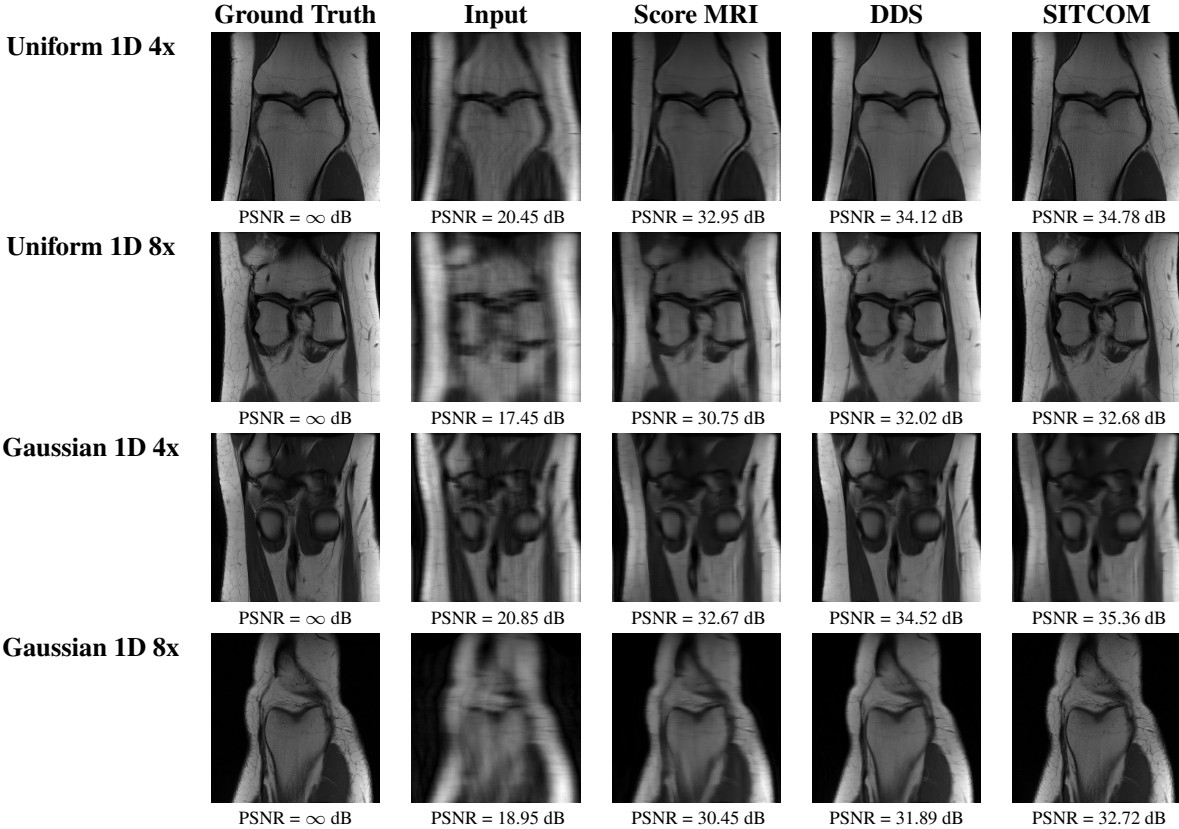

Figure 5: Reconstructed images using our proposed approach, SITCOM, and DM-based baselines (DDS and Score-MRI). Each row corresponds to a different mask pattern and acceleration factor. The ground truth and degraded images are shown in the first and second columns, respectively, followed by the reconstructed imaged from the baselines. The last column presents our method. PSNR results are given at the bottom of each reconstructed image. For all tasks, SITCOM reconstructions contain sharper and clearer image features than other methods.

# I. Additional Comparison Results

## I.1. Near Noiseless Setting

In Table 9, we present the average PSNR, SSIM, LPIPS, and run-time (seconds) of DPS, DAPS, DDNM, and SITCOM using the FFHQ and ImageNet datasets for which the measurement noise level is set to $\sigma_{\mathbf{y}} = 0.01$ (different from Table 1). The goal of these results is to evaluate our method and baselines under less noisy settings.

| Task | Method | FFHQ | | | | ImageNet | | | |
|---|---|---|---|---|---|---|---|---|---|
| | | PSNR (↑) | SSIM (↑) | LPIPS (↓) | **Run-time (↓)** | PSNR (↑) | SSIM (↑) | LPIPS (↓) | **Run-time (↓)** |
| Super Resolution 4× | DPS | $25.20_{\pm1.22}$ | $0.806_{\pm0.044}$ | $0.242_{\pm0.102}$ | $78.60_{\pm26.40}$ | $24.45_{\pm0.89}$ | $0.792_{\pm0.052}$ | $0.331_{\pm0.089}$ | $139.80_{\pm24.00}$ |
| | DAPS | $\underline{29.60}_{\pm0.67}$ | $\underline{0.871}_{\pm0.034}$ | $\mathbf{0.132}_{\pm0.088}$ | $74.40_{\pm25.80}$ | $\underline{25.98}_{\pm0.74}$ | $\underline{0.794}_{\pm0.09}$ | $\underline{0.234}_{\pm0.089}$ | $106.00_{\pm21.20}$ |
| | DDNM | $28.82_{\pm0.67}$ | $0.851_{\pm0.043}$ | $0.188_{\pm0.13}$ | $\underline{64.20}_{\pm21.00}$ | $24.67_{\pm0.78}$ | $0.771_{\pm0.06}$ | $0.432_{\pm0.034}$ | $\underline{82.80}_{\pm33.00}$ |
| | Ours | $\mathbf{30.95}_{\pm0.89}$ | $\mathbf{0.872}_{\pm0.045}$ | $\underline{0.137}_{\pm0.046}$ | $\mathbf{30.00}_{\pm8.40}$ | $\mathbf{26.89}_{\pm0.86}$ | $\mathbf{0.802}_{\pm0.057}$ | $\mathbf{0.224}_{\pm0.056}$ | $\mathbf{80.40}_{\pm17.00}$ |
| Box In-Painting | DPS | $23.56_{\pm0.78}$ | $0.762_{\pm0.034}$ | $0.191_{\pm0.087}$ | $91.20_{\pm25.80}$ | $20.22_{\pm0.67}$ | $0.69_{\pm0.034}$ | $0.297_{\pm0.077}$ | $93.00_{\pm26.40}$ |
| | DAPS | $24.41_{\pm0.67}$ | $\underline{0.791}_{\pm0.034}$ | $\underline{0.129}_{\pm0.067}$ | $69.80_{\pm12.20}$ | $21.79_{\pm0.34}$ | $0.734_{\pm0.045}$ | $\underline{0.214}_{\pm0.034}$ | $96.40_{\pm20.40}$ |
| | DDNM | $\underline{24.67}_{\pm0.067}$ | $0.788_{\pm0.024}$ | $0.229_{\pm0.055}$ | $\underline{61.20}_{\pm15.20}$ | $\underline{21.99}_{\pm0.54}$ | $\underline{0.737}_{\pm0.034}$ | $0.315_{\pm0.022}$ | $\underline{85.20}_{\pm27.00}$ |
| | Ours | $\mathbf{24.97}_{\pm0.55}$ | $\mathbf{0.804}_{\pm0.045}$ | $\mathbf{0.118}_{\pm0.022}$ | $\mathbf{22.20}_{\pm8.40}$ | $\mathbf{22.23}_{\pm0.44}$ | $\mathbf{0.745}_{\pm0.034}$ | $\mathbf{0.208}_{\pm0.023}$ | $\mathbf{73.80}_{\pm26.40}$ |
| Random In-Painting | DPS | $28.77_{\pm0.56}$ | $0.847_{\pm0.034}$ | $0.191_{\pm0.023}$ | $93.00_{\pm20.40}$ | $24.57_{\pm0.45}$ | $0.775_{\pm0.023}$ | $0.318_{\pm0.26}$ | $127.20_{\pm18.00}$ |
| | DAPS | $\underline{31.56}_{\pm0.45}$ | $\mathbf{0.905}_{\pm0.013}$ | $\underline{0.094}_{\pm0.012}$ | $79.20_{\pm17.00}$ | $\underline{28.86}_{\pm0.67}$ | $0.877_{\pm0.021}$ | $\underline{0.131}_{\pm0.044}$ | $120.60_{\pm20.40}$ |
| | DDNM | $30.56_{\pm0.56}$ | $0.902_{\pm0.013}$ | $0.116_{\pm0.023}$ | $\underline{75.00}_{\pm15.20}$ | $30.12_{\pm0.45}$ | $\underline{0.917}_{\pm0.012}$ | $\mathbf{0.124}_{\pm0.032}$ | $\underline{113.40}_{\pm13.80}$ |
| | Ours | $\mathbf{33.02}_{\pm0.44}$ | $\mathbf{0.919}_{\pm0.012}$ | $\mathbf{0.0912}_{\pm0.013}$ | $\mathbf{28.20}_{\pm6.40}$ | $\mathbf{30.67}_{\pm0.45}$ | $\mathbf{0.918}_{\pm0.013}$ | $\underline{0.118}_{\pm0.012}$ | $\mathbf{84.00}_{\pm20.40}$ |
| Gaussian Deblurring | DPS | $25.78_{\pm0.68}$ | $0.831_{\pm0.034}$ | $0.202_{\pm0.014}$ | $79.80_{\pm26.40}$ | $22.45_{\pm0.42}$ | $0.778_{\pm0.067}$ | $0.344_{\pm0.041}$ | $127.20_{\pm26.40}$ |
| | DAPS | $\underline{29.67}_{\pm0.45}$ | $\underline{0.889}_{\pm0.045}$ | $\underline{0.163}_{\pm0.033}$ | $79.00_{\pm22.20}$ | $26.34_{\pm0.55}$ | $0.836_{\pm0.034}$ | $\underline{0.244}_{\pm0.023}$ | $133.20_{\pm25.80}$ |
| | DDNM | $28.56_{\pm0.45}$ | $0.872_{\pm0.024}$ | $0.211_{\pm0.034}$ | $\underline{74.40}_{\pm20.40}$ | $\mathbf{28.44}_{\pm0.021}$ | $\underline{0.882}_{\pm0.021}$ | $0.267_{\pm0.044}$ | $\underline{105.60}_{\pm19.80}$ |
| | Ours | $\mathbf{32.12}_{\pm0.34}$ | $\mathbf{0.913}_{\pm0.024}$ | $\mathbf{0.139}_{\pm0.045}$ | $\mathbf{27.00}_{\pm10.20}$ | $\underline{28.22}_{\pm0.45}$ | $\mathbf{0.891}_{\pm0.014}$ | $\mathbf{0.216}_{\pm0.021}$ | $\mathbf{80.40}_{\pm15.00}$ |
| Motion Deblurring | DPS | $23.78_{\pm0.78}$ | $0.742_{\pm0.042}$ | $0.265_{\pm0.024}$ | $99.00_{\pm20.40}$ | $22.33_{\pm0.727}$ | $0.726_{\pm0.034}$ | $0.352_{\pm0.00}$ | $132.60_{\pm24.00}$ |
| | DAPS | $\underline{30.78}_{\pm0.56}$ | $\underline{0.892}_{\pm0.034}$ | $\underline{0.146}_{\pm0.023}$ | $\underline{66.40}_{\pm20.40}$ | $\underline{28.24}_{\pm0.62}$ | $\underline{0.867}_{\pm0.023}$ | $\underline{0.191}_{\pm0.017}$ | $\underline{127.20}_{\pm26.40}$ |
| | Ours | $\mathbf{32.34}_{\pm0.44}$ | $\mathbf{0.908}_{\pm0.028}$ | $\mathbf{0.135}_{\pm0.028}$ | $\mathbf{31.20}_{\pm10.40}$ | $\mathbf{29.12}_{\pm0.38}$ | $\mathbf{0.882}_{\pm0.025}$ | $\mathbf{0.182}_{\pm0.025}$ | $\mathbf{87.00}_{\pm18.60}$ |
| _Phase Retrieval_ | DPS | $17.56_{\pm2.15}$ | $0.681_{\pm0.056}$ | $0.392_{\pm0.021}$ | $91.20_{\pm25.20}$ | $16.77_{\pm1.78}$ | $0.651_{\pm0.076}$ | $0.442_{\pm0.037}$ | $\underline{130.80}_{\pm22.80}$ |
| | DAPS | $\underline{31.45}_{\pm2.78}$ | $\underline{0.909}_{\pm0.035}$ | $\underline{0.109}_{\pm0.044}$ | $\underline{81.00}_{\pm19.20}$ | $\mathbf{26.12}_{\pm2.12}$ | $\underline{0.802}_{\pm0.043}$ | $\underline{0.247}_{\pm0.044}$ | $139.20_{\pm21.00}$ |
| | Ours | $\mathbf{31.88}_{\pm2.89}$ | $\mathbf{0.921}_{\pm0.067}$ | $\mathbf{0.102}_{\pm0.078}$ | $\mathbf{32.40}_{\pm12.40}$ | $\underline{25.76}_{\pm1.78}$ | $\mathbf{0.813}_{\pm0.032}$ | $\mathbf{0.238}_{\pm0.067}$ | $\mathbf{78.60}_{\pm17.00}$ |
| _Non-Uniform Deblurring_ | DPS | $23.78_{\pm2.23}$ | $0.761_{\pm0.051}$ | $0.269_{\pm0.064}$ | $93.60_{\pm27.00}$ | $22.97_{\pm1.57}$ | $0.781_{\pm0.023}$ | $0.302_{\pm0.089}$ | $140.40_{\pm26.40}$ |
| | DAPS | $\underline{28.89}_{\pm1.67}$ | $\underline{0.845}_{\pm0.057}$ | $\underline{0.150}_{\pm0.056}$ | $\underline{84.60}_{\pm22.20}$ | $\underline{28.02}_{\pm1.15}$ | $\underline{0.831}_{\pm0.082}$ | $\underline{0.162}_{\pm0.034}$ | $\underline{133.80}_{\pm33.60}$ |
| | Ours | $\mathbf{31.09}_{\pm0.89}$ | $\mathbf{0.911}_{\pm0.056}$ | $\mathbf{0.132}_{\pm0.45}$ | $\mathbf{33.60}_{\pm8.20}$ | $\mathbf{29.56}_{\pm0.78}$ | $\mathbf{0.844}_{\pm0.045}$ | $\mathbf{0.147}_{\pm0.042}$ | $\mathbf{80.40}_{\pm16.40}$ |
| _High Dynamic Range_ | DPS | $23.33_{\pm1.34}$ | $0.734_{\pm0.049}$ | $0.251_{\pm0.078}$ | $80.40_{\pm25.20}$ | $19.67_{\pm0.056}$ | $0.693_{\pm0.034}$ | $0.498_{\pm0.112}$ | $140.40_{\pm24.60}$ |
| | DAPS | $\underline{27.58}_{\pm0.829}$ | $\underline{0.828}_{\pm0.00}$ | $\underline{0.161}_{\pm0.067}$ | $\underline{75.60}_{\pm16.40}$ | $\underline{26.71}_{\pm0.088}$ | $\underline{0.802}_{\pm0.032}$ | $\underline{0.172}_{\pm0.066}$ | $\underline{127.20}_{\pm19.20}$ |
| | Ours | $\mathbf{28.52}_{\pm0.89}$ | $\mathbf{0.844}_{\pm0.045}$ | $\mathbf{0.148}_{\pm0.035}$ | $\mathbf{30.60}_{\pm7.20}$ | $\mathbf{27.56}_{\pm0.78}$ | $\mathbf{0.825}_{\pm0.037}$ | $\mathbf{0.162}_{\pm0.046}$ | $\mathbf{87.00}_{\pm14.60}$ |

Table 9: Average PSNR, SSIM, LPIPS, and **run-time (seconds)** of SITCOM and baselines using 100 test images from FFHQ and 100 test images from ImageNet with a **measurement noise level** of $\sigma_{\mathbf{y}} = 0.01$. The first five tasks are linear, while the last three tasks are non-linear (underlined). For each task and dataset combination, the best results are in bold, and the second-best results are underlined. Values after ± represent the standard deviation. All results were obtained using a **single RTX5000 GPU** machine. For phase retrieval, the run-time is reported for the best result out of four independent runs. This is applied for SITCOM and baselines.

Overall, we observe similar trends to those discussed in Section 4 for Table 1. On the FFHQ dataset, SITCOM achieves higher average PSNR values compared to the baselines across all tasks, with improvements exceeding 1 dB in 5 out of 8 tasks. For the ImageNet dataset, we observe more than 1 dB improvement on the non-linear deblurring task, while for the remaining tasks, the improvement is less than 1 dB, except for Gaussian deblurring (where SITCOM underperforms by 0.22 dB) and phase retrieval (underperforming by 0.36 dB).

In terms of run-time, generally, SITCOM outperforms DDNM, DPS, and DAPS, with all methods evaluated on a single RTX5000 GPU. For the FFHQ dataset, SITCOM is at least twice as fast when compared to baselines. On ImageNet, SITCOM consistently requires less run-time compared to DPS and DAPS. When compared to DDNM, SITCOM's run-time is similar or slightly lower. For example, on the super-resolution task on ImageNet, both SITCOM and DDNM average around 81 seconds, but SITCOM achieves over a 2 dB improvement.

## I.2. Performance Comparison Under Time Constraints

In this subsection, we compare the performance of SITCOM with baselines where their default settings require less run-time than SITCOM. Namely, DCDP on linear IPs, RED-diff, and PGDM. In Section 4, we demonstrated that the proposed step-wise backward consistency significantly reduces the number of sampling steps. However, enforcing backward consistency also increases the runtime of each iteration, as the gradient computation in SITCOM requires backpropagation through the

neural network. Here, we assess the combined impact of these two factors—fewer sampling steps and increased runtime per step—by conducting time-budgeted comparisons with fast baselines. We will compare SITCOM with DCDP, which currently achieves SOTA runtime performance, as well as with RED-diff and PGDM, which enforce (non-stepwise) network consistency.

For a fair comparison, we evaluate the runtime of each method in relation to achieving the same PSNR.

Table 10 includes a comparison of SITCOM and DCDP. DCDP has two versions, and we only compare with its version 1 as it delivers notably better results than version 2. The parameters $N$ and $K$ in SITCOM also appear in DCDP, so we report results of the two methods for various values of $N$ and $K$. DCDP has an extra inner for-loop (total of 3 nested-for-loop in DCDP) which applies diffusion purification. We fixed the time schedule for the diffusion purification to be the default one. In addition, we set $\sigma_{\mathbf{y}} = 0$ because SITCOM and DCDP handle measurement noise differently.

| Method | $N$ | $K$ | PSNR | Run-time (seconds) |
|---|---|---|---|---|
| SITCOM | 10 | 10 | 31.6 | 8.72 |
| DCDP (Default) | 10 | 10 | 29.52 | 10.50 |
| SITCOM (Default) | 20 | 20 | 32.28 | 28.12 |
| DCDP | 20 | 20 | 31.67 | 16.24 |
| SITCOM | 10 | 30 | 31.66 | 22.11 |
| DCDP | 10 | 30 | 31.11 | 12.42 |
| SITCOM | 30 | 10 | 31.34 | 22.08 |
| DCDP | 30 | 10 | 30.81 | 12.04 |
| SITCOM | 50 | 10 | 32.32 | 32.22 |
| DCDP | 50 | 10 | 31.69 | 26.20 |
| SITCOM | 100 | 10 | 32.34 | 51.23 |
| DCDP | 100 | 10 | 31.71 | 42.45 |
| SITCOM | 200 | 10 | 32.41 | 124.20 |
| DCDP | 200 | 10 | 31.76 | 112.10 |
| SITCOM | 500 | 10 | 32.46 | 320.20 |
| DCDP | 500 | 10 | 31.82 | 309.10 |

Table 10: Comparison of SITCOM vs. DCDP using the linear inverse problem, Gaussian Deblurring (we consider linear problems here since SITCOM already demonstrates a significant advantage in run-time for nonlinear settings, as evidenced in Table 1 of the main text). Results are reported using the FFHQ dataset with $\sigma_{\mathbf{y}} = 0$. Different values of $N$ and $K$ are used to report the average PSNR and run-time (seconds).

As confirmed by the table, SITCOM is slower than DCDP (which does not enforce the backward consistency) when the number of iterations is fixed due to the need to back-propagate. However, SITCOM allows a more significant reduction of $N$ and $K$ while achieving competitive PSNRs, so the total run time of SITCOM is smaller. For instance, the PSNR of SITCOM with $N = 10$ matches DCDP with $N = 20$.

In Table 11 and Table 12, we compare SITCOM with RED-diff and PGDM, respectively. Both of these baselines utilize ideas related to network regularization. However, they utilize the entire diffusion process (from time $T$ to 0) as a single regularizer of the reconstructed image, instead of applying the network regularize stepwise. Therefore, they're not imposing the step-wise backward consistency.

| Method | Super Resolution | Motion Deblurring | High Dynamic Range |
|---|---|---|---|
| RED-diff | 24.89/18.02 | 27.35/19.02 | 23.45/24.40 |
| SITCOM (Default) | 26.35/62.02 | 28.65/62.04 | 26.97/62.50 |
| SITCOM $(N, K) = (8, 8)$ | 24.99/17.80 | 27.38/17.92 | 25.24/20.12 |

Table 11: Average PSNR/rune-time of SITCOM as compared to RED-diff using two linear tasks and one non-linear task. Here, ImageNet is used with noise level of $\sigma_{\mathbf{y}} = 0.05$.

As observed in Table 11, for Super Resolution and Motion Deblurring, the default SITCOM (second line) requires more run-time and achieves better PSNR than RED-diff. However, the third line shows that if we just want to match the performance

of RED-diff, we can use a smaller $N$ and $K$ for SITCOM that greatly reduces the run-time to below RED-diff.

| Method | Super Resolution | Motion Deblurring |
|---|---|---|
| PGDM | 25.22/26.20 | 27.48/24.10 |
| SITCOM (Default) | 26.35/62.02 | 28.65/62.04 |
| SITCOM $(N, K) = (10, 10)$ | 25.15/19.72 | 27.55/19.65 |

Table 12: Average PSNR/rune-time of SITCOM as compared to PGDM for super resolution and motion deblurring. Here, ImageNet is used with noise level of $\sigma_{\mathbf{y}} = 0.05$.

As observed in Table 12, given the same PSNR, SITCOM requires less run-time than PGDM. Additionally, the default $N$ and $K$ allow SITCOM to achieve more than 1dB improvement.

## J. Ablation Studies

### J.1. Impact of $K$ & $N$ and Convergence Plots

In this subsection, we perform an ablation study on the number of optimization steps, $K$, and the number of sampling steps, $N$. Specifically, for the task of Gaussian Deblurring, we run SITCOM using combinations from $N \in \{4, 8, 12, \dots, 40\}$ and $K \in \{5, 10, 15, \dots, 40\}$. The average PSNR results over 20 test images from the FFHQ dataset are presented in Table 13.

We observe that with $N = 12$ and $K = 15$, we obtain 28.3dB (with only 18.3 seconds) whereas at $N = K = 40$, we obtain 29 dB which required nearly 110 seconds. This indicates that there is indeed a trade-off between computational cost and PSNR values. Furthermore, we observe that 61 entries (out of 80) achieve a PSNR values of more than or equal to 28 dB. This indicates that *SITCOM is not very sensitive to the choice* of $N$ and $K$. We note that we observe similar patterns for all tasks. See the tables of all the tasks here[6].

The selected $(N, K)$ values for our main results are listed in Table 17 of Appendix K.1.

| $(N, K)$ | $K = 5$ | $K = 10$ | $K = 15$ | $K = 20$ | $K = 25$ | $K = 30$ | $K = 35$ | $K = 40$ |
|---|---|---|---|---|---|---|---|---|
| $N = 4$ | 19.3, 5.22 | 23.6, 7.14 | 25.9, 9.00 | 26.7, 10.56 | 27.3, 11.34 | 27.7, 12.48 | 27.9, 13.74 | 28.1, 12.96 |
| $N = 8$ | 23.7, 7.32 | 27.2, 11.04 | 27.9, 13.98 | 28.3, 17.16 | 28.4, 19.80 | 28.5, 22.98 | 28.5, 25.32 | 28.6, 23.76 |
| $N = 12$ | 25.2, 9.42 | 27.9, 14.28 | 28.3, 18.30 | 28.5, 22.26 | 28.6, 26.76 | 28.7, 30.54 | 28.8, 35.94 | 28.7, 34.50 |
| $N = 16$ | 26.2, 10.92 | 28.2, 17.58 | 28.5, 23.70 | 28.6, 29.76 | 28.7, 36.18 | 28.8, 42.18 | 28.8, 47.82 | 28.8, 44.82 |
| $N = 20$ | 26.7, 13.02 | 28.4, 21.12 | 28.6, 28.86 | 28.7, 36.54 | 28.8, 43.98 | 28.8, 51.12 | 28.9, 57.96 | 28.9, 54.72 |
| $N = 24$ | 27.1, 14.58 | 28.4, 24.36 | 28.7, 33.96 | 28.7, 43.20 | 28.9, 52.02 | 28.8, 60.48 | 28.9, 98.04 | 28.8, 65.10 |
| $N = 28$ | 27.3, 23.22 | 28.5, 39.72 | 28.7, 56.58 | 28.8, 72.18 | 28.9, 88.38 | 28.9, 103.44 | 29.0, 117.90 | 28.9, 75.54 |
| $N = 32$ | 27.6, 25.62 | 28.5, 45.00 | 28.7, 64.14 | 28.8, 82.86 | 28.9, 98.34 | 28.9, 114.84 | 28.9, 123.96 | 29.0, 86.64 |
| $N = 36$ | 27.8, 24.66 | 28.7, 44.46 | 28.8, 61.44 | 28.9, 74.16 | 28.9, 84.78 | 29.0, 89.04 | 29.0, 99.78 | 29.0, 96.12 |
| $N = 40$ | 27.8, 21.72 | 28.6, 38.04 | 28.9, 53.82 | 28.9, 69.06 | 29.0, 84.06 | 29.0, 99.48 | 29.0, 121.62 | 29.0, 109.80 |

Table 13: Performance comparison for different $(N, K)$ using the task of Gaussian Deblurring. Each cell shows values in the format PSNR, Run-Time (seconds).

Figure 6 shows PSNR curves of SITCOM using different values of $N$ and $K$ for Gaussian Deblurring and High Dynamic Range. As observed, with $NK = 400$, $(N, K) = (10, 40)$ yields the best results. Notably, after completing the optimization steps at each sampling step, we observe a drop in PSNR, attributed to the application of the resampling formula in ($S_3$) where additive noise is added to the estimated image at each sampling discrete time step.

### J.2. Ablation Studies on the Stopping Criterion For Noisy Measurements

In SITCOM, we proposed to use a stopping criterion to prevent the noise overfitting, determined by $\sigma_{\mathbf{y}}$. In this subsection, we (*i*) show how sensitive SITCOM's performance is w.r.t. the choice of $\delta$, and (*ii*) we visually show the impact of using

---

[6] https://github.com/sjames40/SITCOM/blob/main/Extended_Ablation.md

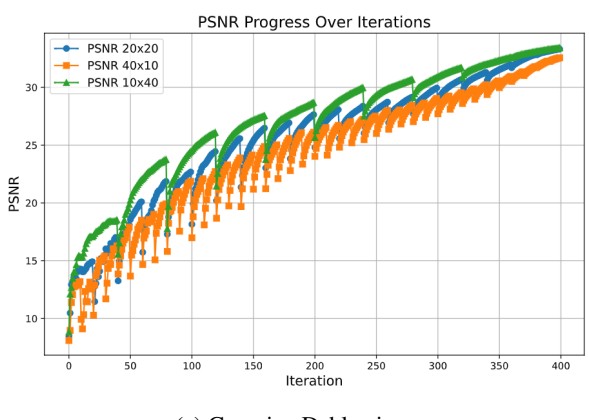

(a) Gaussian Deblurring.

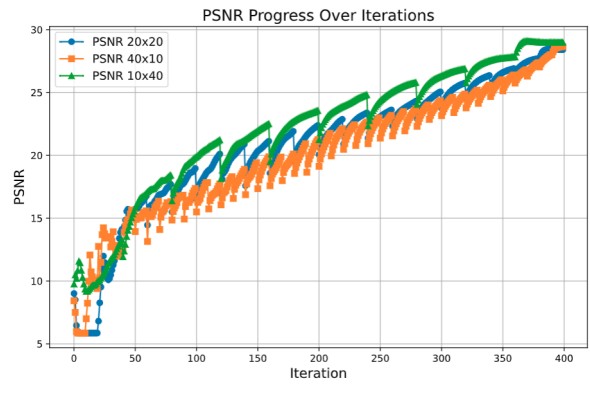

(b) High Dynamic Range.

Figure 6: SITCOM PSNR curves for Gaussian Deblurring (left) and High Dynamic Range (right) tasks using different values of $N$ and $K$ (labeled as $N \times K$ in the legends).

the stopping criterion vs. not using it; (*iii*) We evaluate SITCOM using measurement noise sourced from a non-Gaussian distribution.

### J.2.1. SENSITIVITY TO THE CHOICE OF THE STOPPING CRITERION $\delta$

In SITCOM, we proposed to use a stopping criterion to prevent the noise overfitting, determined by $\sigma_{\mathbf{y}}$. Here, we try to answer the question: *How sensitive is SITCOM's performance to the choice of the stopping criterion parameter $\delta$?* To answer this question, we run SITCOM using different values of $\delta$ and report the average PSNR values and run-time (in seconds).

We use $N = K = 20$ and $\sigma_{\mathbf{y}} = 0.05$ and set $\delta$ to values above and below $\sigma_{\mathbf{y}}$. Results for Super-Resolution(linear IP) and Non-uniform deblurring (non-linear IP) are given in Table 14, where we use 20 FFHQ test images with $m = 256 \times 256 \times 3$. The first row shows the values of $a$ for which $\delta = a\sqrt{m}$.

| Task | 0.01 | 0.017 | 0.025 | 0.04 | 0.05 | 0.051 | 0.055 | 0.07 | 0.085 | 0.1 | 0.5 |
|------|------|-------|-------|------|------|-------|-------|------|-------|-----|-----|
| SR | 27.80/40.20 | 28.45/38.82 | 29.02/38.41 | 30.22/36.26 | 30.39/28.52 | 30.40/28.26 | 30.37/24.10 | 30.11/23.5 | 29.87/22.89 | 28.87/22.12 | 23.10/15.87 |
| NDB | 28.40/40.40 | 28.87/38.58 | 29.80/37.32 | 29.82/35.12 | 30.12/32.12 | 30.21/29.46 | 30.20/27.44 | 30.01/26.56 | 29.67/24.52 | 29.12/22.05 | 25.79/15.10 |

Table 14: Performance comparison for Super Resolution (SR) and Non-linear Deblurring (NDB) of different values of $a$ (first row) for which $\delta = a\sqrt{m}$ and $\sigma_{\mathbf{y}} = 0.05$. Each cell shows values in the format PSNR/run-time (seconds).

As observed, the PSNR values vary between 29.02 to 30.37 (resp. 29.80 to 30.20) only for BIP (resp. NDB) with stopping criterion between 0.025 to 0.055 which indicates that even if values lower (or slightly higher) than the measurement noise level are selected, SITCOM can still perform very well. This means that if we use classical methods (such as (Liu et al., 2006; Chen et al., 2015)) to approximate/estimate the noise, we can achieve competitive results.

We note that other works, such as DAPS (Zhang et al., 2025) and PGDM (Song et al., 2023b), also use $\sigma_{\mathbf{y}}$ but not to estimate the stopping criteria. In these papers, $\sigma_{\mathbf{y}}$ is encoded in the updates of their algorithm (see Eq. (7) in PGDM and Eq. (9) in DAPS).

### J.2.2. IMPACT OF USING THE STOPPING CRITERION VS. NOT USING IT VISUALLY

In this subsection, we demonstrate the impact of applying the stopping criterion in SITCOM when handling measurement noise. For the tasks of super resolution and motion deblurring, we run SITCOM with and without the stopping criterion for the case of $\sigma_{\mathbf{y}} = 0.05$. The results are presented in Figure 7. As shown, for both tasks, using the stopping criterion (i.e., $\delta > 0$) not only improves PSNR values compared to the case of $\delta = 0$, but also visually reduces additive noise in the restored images. This is because, without the stopping criterion, the measurement consistency enforced by the optimization in ($S_1$) tends to fit the noise in the measurements.

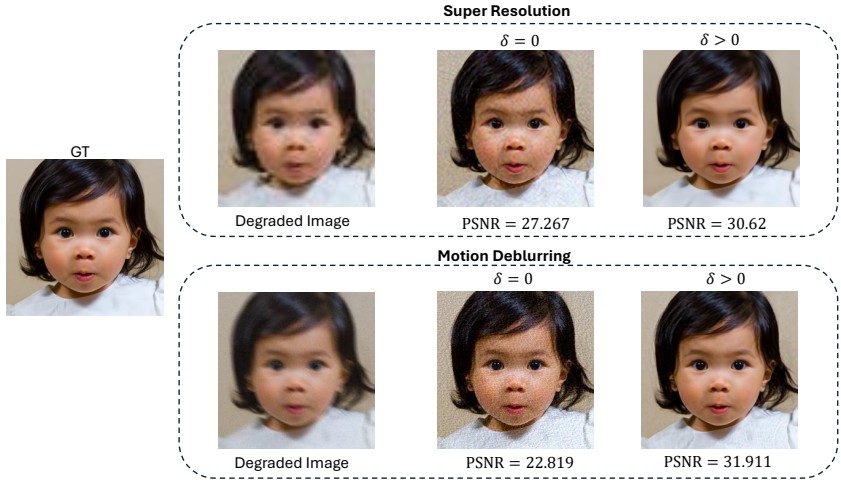

Figure 7: Impact of the stopping criterion in preventing noise overfitting. For the most right column, $\delta$ is set as in Table 17.

### J.2.3. HANDLING NON-GAUSSIAN MEASUREMENT NOISE

Here, we evaluate SITCOM with additive measurement noise vector sampled from the Poisson distribution. We use $N = K = 20$ and $\lambda_y = 5$ (which determines the noise level). We set $\delta$ (the stopping criterion) to different values (top row). Results for Super Resolution (linear IP) and Non-uniform deblurring (non-linear IP) are given in Table 15 where we use 20 FFHQ test images with $m = 256 \times 256 \times 3$. The first row shows the values of $a$ for which $\delta = a\sqrt{m}$.

| Task | 0 | 0.02 | 0.035 | 0.05 | 0.06 | 0.061 | 0.065 | 0.08 | 0.1 | 0.5 |
|------|---|------|-------|------|------|-------|-------|------|-----|-----|
| SR | 24.52/47.30 | 27.80/40.19 | 29.19/38.11 | 30.42/36.23 | 30.49/28.22 | 30.43/28.21 | 30.37/24.16 | 29.67/24.32 | 26.77/22.08 | 25.12/15.88 |
| NDB | 25.04/43.42 | 28.60/40.34 | 30.17/37.52 | 30.22/35.44 | 30.24/32.02 | 30.23/29.31 | 30.15/27.51 | 29.71/24.02 | 29.03/22.25 | 26.44/15.15 |

Table 15: Average PSNR/run-time (seconds) of SITCOM for the tasks of Super-Resolution (SR) and Non-uniform Deblurring (NDB) across various stopping criterion thresholds (top row is the value of $a$ for which $\delta = a\sqrt{m}$)) using **additive noise from the Poisson distribution** (code from DPS [7])

As observed, the results of setting $\delta$ to values between $0.035\sqrt{m}$ to $0.08\sqrt{m}$ return PSNR values within approximately 1 dB. This indicates that SITCOM can perform reasonably well with different values of the stopping criterion even if its settings was designed for the additive Gaussian measurement noise.

### J.3. Effect of the Regularization Parameter $\lambda$

In this subsection, we perform an ablation study to assess the impact of the regularization parameter, $\lambda$, in SITCOM. Table 16 shows the results across four tasks using various $\lambda$ values. Aside from phase retrieval, the effect of $\lambda$ is minimal. We hypothesize that initializing the optimization variable in ($S_1$) with $\mathbf{x}_t$ is sufficient to enforce forward diffusion consistency in **C3**. Therefore, we set $\lambda = 1$ for phase retrieval and $\lambda = 0$ for the other tasks. The impact of $\lambda$ for all tasks are online[8].

Additionally, for all tasks other than phase retrieval, we observed that when $\lambda = 0$, the restored images exhibit enhanced high-frequency details. For visual examples, see the results of $\lambda = 0$ versus $\lambda = 1$ in Figure 8.

| Task | $\lambda = 0$ | $\lambda = 0.05$ | $\lambda = 0.5$ | $\lambda = 1$ | $\lambda = 1.5$ |
|------|---------------|------------------|-----------------|---------------|-----------------|
| Super Resolution 4× | 29.952 | 29.968 | 29.464 | 29.550 | 29.288 |
| Motion Deblurring | 31.380 | 31.393 | 31.429 | 31.382 | 31.150 |
| Random Inpainting | 34.559 | 34.537 | 34.523 | 34.500 | 34.301 |
| Phase Retrieval | 31.678 | 31.892 | 32.221 | 32.342 | 32.124 |

Table 16: Impact of the regularization parameter $\lambda$ in terms of PSNR for four tasks.

---

[8] https://anonymous.4open.science/r/SITCOM-7539/Extended_Ablation.md

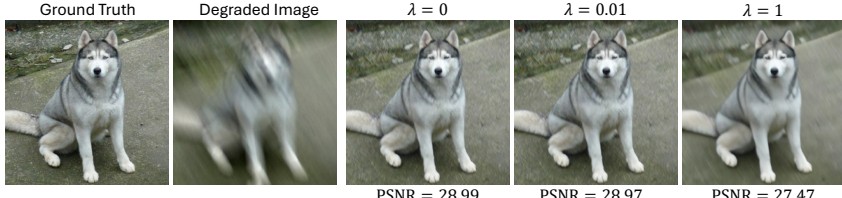

Figure 8: Results of running SITCOM using different regularization parameters in ($S_1$) for the task of Motion deblurring.

## K. Detailed Implementation of Tasks, Baselines, and Hyper-parameters

Our experimental setup for image restoration problems and noise levels largely follows DPS (Chung et al., 2023b). For linear IPs, we evaluate five tasks: super resolution, Gaussian deblurring, motion deblurring, box inpainting, and random inpainting. For Gaussian deblurring and motion deblurring, we use 61×61 kernels with standard deviations of 3 and 0.5, respectively. In the super-resolution task, a bicubic resizer downscales images by a factor of 4. For box inpainting, a random 128×128 box is applied to mask image pixels, and for random inpainting, the mask is generated with each pixel masked with a probability of 0.7, as described in (Song et al., 2023a).

For nonlinear IP tasks, we consider three tasks: phase retrieval, high dynamic range (HDR) reconstruction, and nonlinear (non-uniform) deblurring. In HDR reconstruction, the goal is to restore a higher dynamic range image from a lower dynamic range image (with a factor of 2). Nonlinear deblurring follows the setup in (Tran et al., 2021).

For MRI, we follow the setup in Decomposed Diffusion Sampling (DDS) (Chung et al., 2024) for which we consider different mask patterns and acceleration factors (Ax).

**Baselines & Datasets:** For baselines, we use DPS (Chung et al., 2023b), DDNM (Wang et al., 2022), RED-diff (Mardani et al., 2024), PGDM (Song et al., 2023b), DCDP (Li et al., 2024), DMPlug (Wang et al., 2024), and DAPS (Zhang et al., 2025). The selection criteria is based on these baselines' competitive performance on several linear and non-linear inverse problems under measurement noise. For MRI, we compare against diffusion-based solvers (DDS (Chung et al., 2024) and Score-MRI (Chung & Ye, 2022)), supervised learning solvers (Supervised U-Net(Ronneberger et al., 2015) and E2E Varnet(Sriram et al., 2020)), and dataless approaches (Self-Guided DIP(Liang et al., 2025), Auto-encoded Sequential DIP (aSeqDIP)(Alkhouri et al., 2024a), and TV-ADMM (Uecker et al., 2014)). We evaluate SITCOM and baselines using 100 test images from the validation set of FFHQ (Karras et al., 2019) and 100 test images from the validation set of ImageNet (Deng et al., 2009) for which the FFHQ-trained and ImageNet-trained DMs are given in (Chung et al., 2023b) and (Dhariwal & Nichol, 2021), respectively, following the previous convention.

For baselines, we used the codes provided by the authors of each paper: DPS[9], DDNM[10], DAPS[11], and DCDP[12]. Default configurations are used for each task.

### K.1. Complete List of hyper-parameters in SITCOM

Table 17 summarizes the hyper-parameters used for each task in our experiments, as determined by the ablation studies in the previous subsections. Notably, the same set of hyper-parameters is applied to both the FFHQ and ImageNet datasets.

## L. Additional Qualitative results

Figure 9 (resp. 10) presents results with SITCOM, DPS, and DAPS using FFHQ (resp. ImageNet). See also Figure 11, Figure 12, Figure 13, and Figure 14 for more images.

---

[9] https://github.com/DPS2022/diffusion-posterior-sampling
[10] https://github.com/wyhuai/DDNM
[11] https://github.com/zhangbingliang2019/DAPS
[12] https://github.com/Morefre/Decoupled-Data-Consistency-with-Diffusion-Purification-for-Image-Restoration

| Task | Sampling Steps $N$ | Optimization Steps $K$ | Regularization $\lambda$ | Stopping criterion $\delta$ for $\sigma_{\mathbf{y}} \in \{0.05, 0.01\}$ |
|---|---|---|---|---|
| Super Resolution $4\times$ | 20 | 20 | 0 | $\{0.051\sqrt{m_{\mathrm{SR}}}, 0.011\sqrt{m_{\mathrm{SR}}}\}$ |
| Box In-Painting | 20 | 20 | 0 | $\{0.051\sqrt{m}, 0.011\sqrt{m}\}$ |
| Random In-Painting | 20 | 30 | 0 | $\{0.051\sqrt{m}, 0.011\sqrt{m}\}$ |
| Gaussian Deblurring | 20 | 30 | 0 | $\{0.051\sqrt{m}, 0.011\sqrt{m}\}$ |
| Motion Deblurring | 20 | 30 | 0 | $\{0.051\sqrt{m}, 0.011\sqrt{m}\}$ |
| Phase Retrieval | 20 | 30 | 1 | $\{0.051\sqrt{m_{\mathrm{PR}}}, 0.011\sqrt{m_{\mathrm{PR}}}\}$ |
| Non-Uniform Deblurring | 20 | 30 | 0 | $\{0.051\sqrt{m}, 0.011\sqrt{m}\}$ |
| High Dynamic Range | 20 | 40 | 0 | $\{0.051\sqrt{m}, 0.011\sqrt{m}\}$ |

Table 17: Hyper-parameters of SITCOM for every image restoration tasks considered in this paper. The same set of hyper-parameters is used for FFHQ and ImageNet. The learning rate in Algorithm 1 is set to $\gamma = 0.01$ for all tasks, datasets, and measurement noise levels. For the stopping criterion column, $m_{\mathrm{SR}} = 64 \times 64 \times 3$, $m = 256 \times 256 \times 3$, and $m_{\mathrm{PR}} = 384 \times 384 \times 3$.

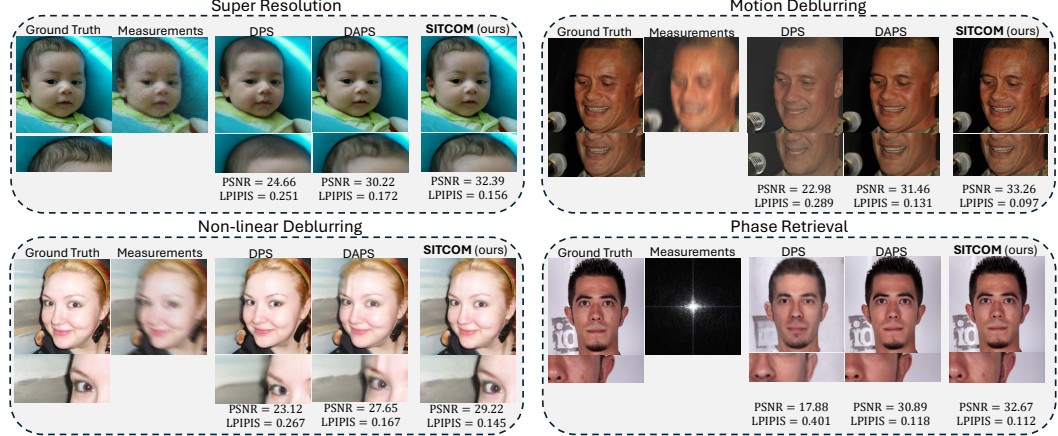

Figure 9: **Qualitative results on the FFHQ dataset** on two linear tasks (*top*) and two non-linear tasks (*bottom*) under measurement noise of $\sigma_{\mathbf{y}} = 0.05$. The PSNR and LPIPS values are given below each restored image. Zoomed-in regions show how SITCOM captures greater image details when compared to two general (non)linear DM-based methods (DPS (Chung et al., 2023b) and DAPS (Zhang et al., 2025)).

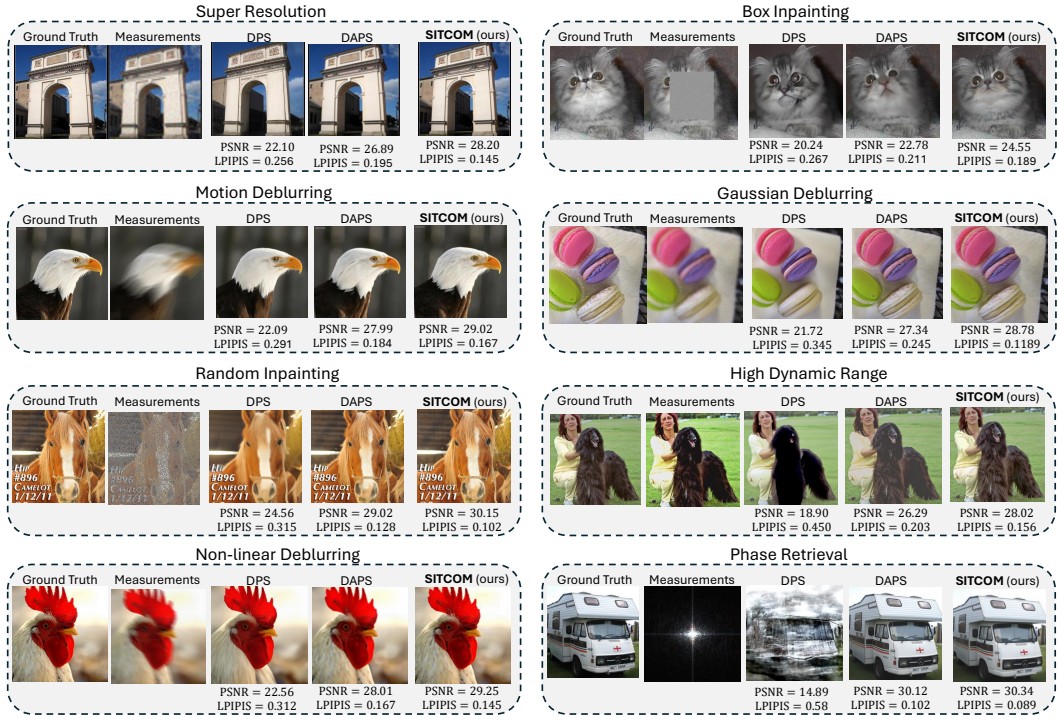

Figure 10: **Qualitative results on the ImageNet dataset** for five linear tasks and three non-linear tasks under measurement noise of $\sigma_{\mathbf{y}} = 0.05$. The PSNR and LPIPS values are given below each restored image.

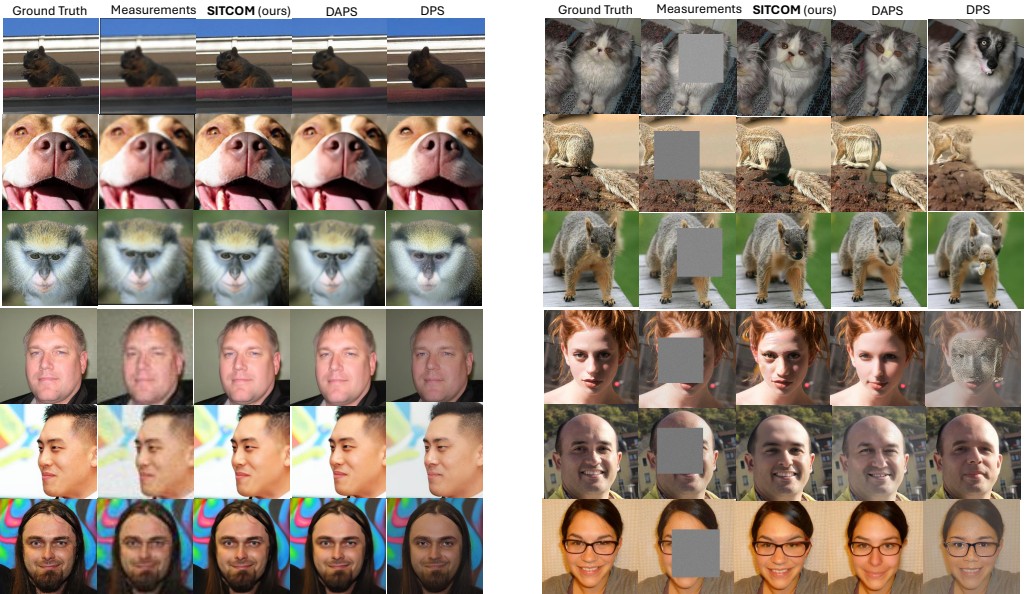

Figure 11: **Super resolution** (*left*) and **box inpainting** (*right*) results. First (resp. last) three rows are for the FFHQ (resp. ImageNet) dataset.

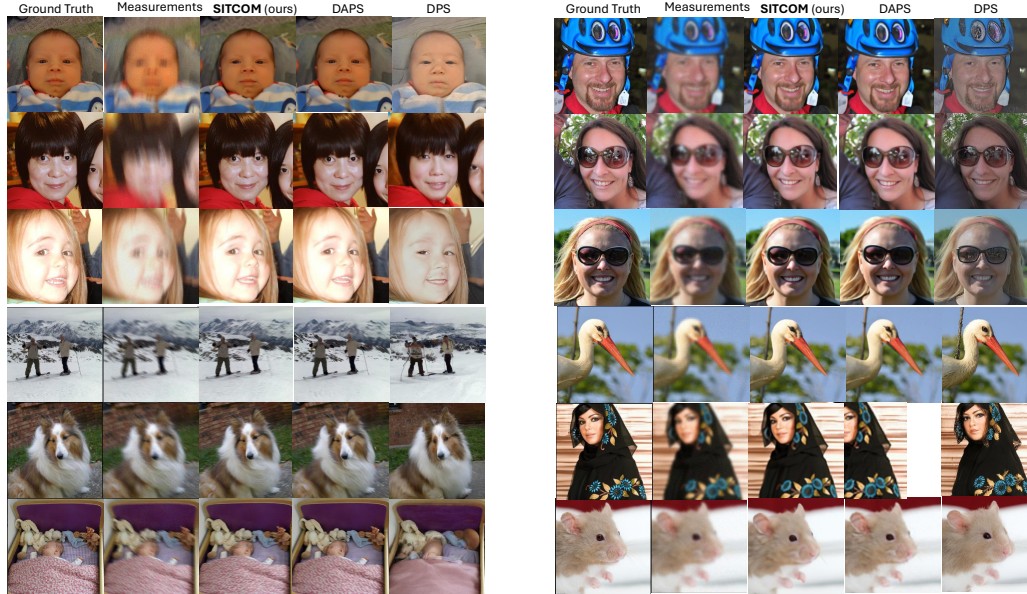

Figure 12: **Motion deblurring** (*left*) and **Gaussian deblurring** (*right*) results. First (resp. last) three rows are for the FFHQ (resp. ImageNet) dataset.

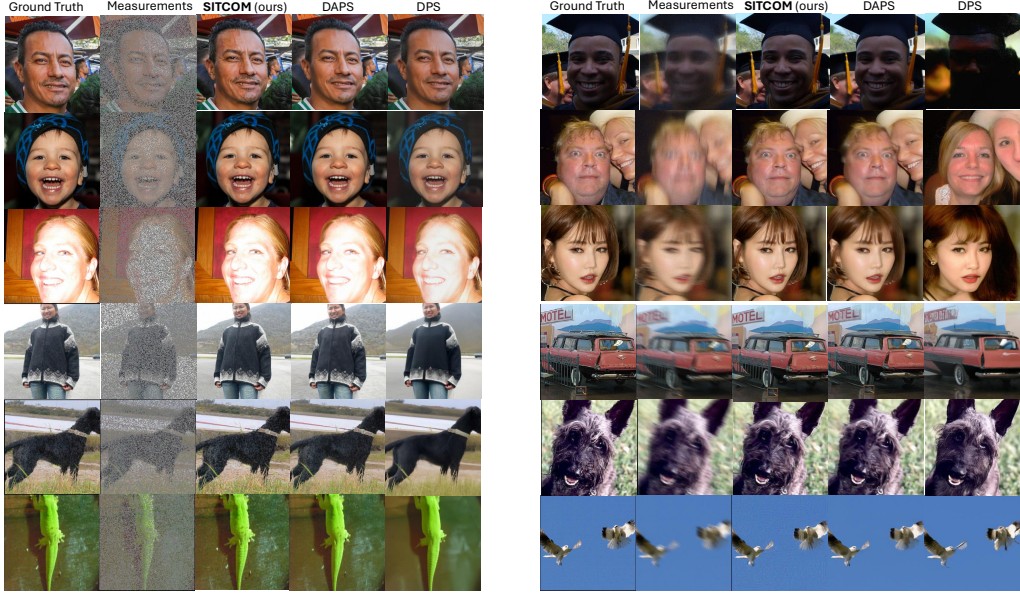

Figure 13: **Random inpainting** (*left*) and **non-linear (non-uniform) deblurring** (*right*) results. First (resp. last) three rows are for the FFHQ (resp. ImageNet) dataset.

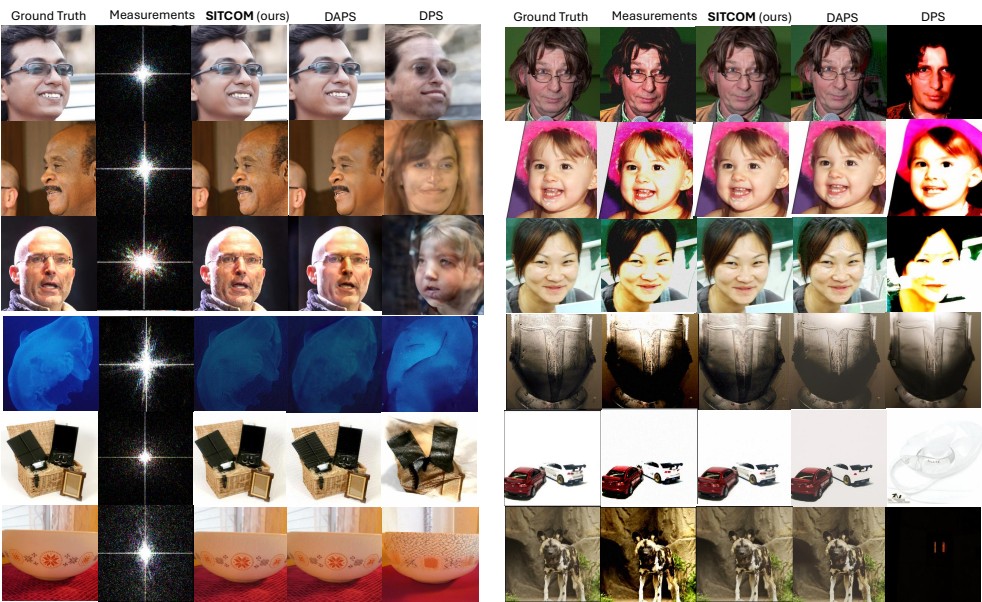

Figure 14: **Phase retrieval** (*left*) and **high dynamic range** (*right*) results. First (resp. last) three rows are for the FFHQ (resp. ImageNet) dataset.

