# OpenReview forum: "SITCOM: Step-wise Triple-Consistent Diffusion Sampling For Inverse Problems"
_ICML.cc/2025/Conference — ICML 2025 poster_

### Official Review · Reviewer_Q3J7 · 2025-03-14

**Overall Recommendation:** 4

**Summary:**

Authors propose Step-wise Triple-Consistent Sampling (SITCOM) an optimization-based sampling method that enforces measurement consistency, forward diffusion consistency, and incorporates step-wise and network-regularized backward diffusion consistency that maintains a diffusion trajectory by optimizing over the input of the pre-trained model at every sampling step. Authors show that these conditions require significantly less number of reverse diffusion steps. Authors demonstrate the efficacy of their method for different levels of measurement noise, various operators (five linear and three nonlinear image restoration tasks and one medical image reconstruction) and show that SITCOM achieves competitive or superior results in terms of standard image similarity metrics and required run-time.

**Claims And Evidence:**

Claims made in the submission are supported by clear and convincing evidence.

**Essential References Not Discussed:**

No.

**Experimental Designs Or Analyses:**

Experimental design follow what is standard in multitude of works (DPS, ReSample, etc.). Analysis of experiments are sound.

**Methods And Evaluation Criteria:**

Selected baselines, tasks, and the datasets are all relevant and appropriate for the application at hand.

**Other Comments Or Suggestions:**

***Suggestions:***
* I would expect increased peak GPU memory usage when back-propagating through the denoising model. This should be disclosed as a limitation.
* I would recommend revising the SITCOM algorithm visualization in Figure 2 (left). The arrows at the bottom guide the reader to look from left to right whereas it is the opposite. Moreover, left and right figures are too close to each other which makes it look visually crowded.

**Other Strengths And Weaknesses:**

***Strengths:***
* Experimental results in terms of baselines and tasks are diverse, extensive and convincing.
* Paper is written very well. I enjoyed reading it.
* Extensive ablation studies are conducted. These include noise robustness, choice of consistency terms, choice of $N$ and $K$, stopping criteria.

***
***Weaknesses:***
* The paper combines existing ideas from ReSample, DPS and DCDP. In that sense, technical novelty is limited.
* The method is inherently more computation heavy due to the need for back-propagating through the denoising model. While the runtime can be offset with smaller choice of $N$ and $K$, it will be more memory hungry compared to baseline models that do not back-propagate through the model.

**Questions For Authors:**

1) Could you clarify the range of images before passing it to the measurement operator? [0,1], [-1,1], etc.?

**Relation To Broader Scientific Literature:**

The core ideas follow from the papers ReSample, DPS, and DCDP. This work generalizes ReSample method which is extensively discussed in Appendix D.1 by the authors.

**Theoretical Claims:**

The paper does not have theoretical claims. The derivations of equation 5 (Appendix A) appears correct to me.Experimental design follow what is standard in multitute of works (DPS, ReSample, etc.). Analysis of experiments are sound.

---

> ### Author Rebuttal · Authors · 2025-04-01
>
> We thank the reviewer for their comments.
>
> We appreciate the reviewer’s recognition of our clear and well-supported claims, as well as their positive feedback on our experimental analysis and writing.
>
> Our point-by-point response is provided below.
>
> ### (W2 and S1) **Memory usage for SITCOM.**
>
> Thank you for the comment! Yes, in the paper we only discussed computation load instead of memory usage. Indeed, due to the inclusion of back-propagation, we require the output of each intermediate layer to be stored, increasing the memory usage from $\mathcal{O}(C\times H\times W)$ to $\mathcal{O}(L\times C\times H\times W)$, where $C$, $H$, $W$ are the maximum number of channels, maximum image height and width across outputs of all layers, and $L$ is the total number of layers.
>
> As a result, our memory usage is approximately $L$ times higher than that of methods requiring only a forward pass, assuming that the output sizes are roughly the same across all layers. However, **the specific Unet architecture** currently used in the diffusion model has an unbalanced layer structure**, where each layer output in the encoder is roughly half the size of the previous one, and each layer output in the decoder is approximately twice the size of the previous one. Consequently, the peak memory consumption due to storing all intermediate outputs is only moderately increased, **roughly four times that of forward-only propagation**.
>
> In addition, **it is possible to greatly reduce the memory by incorporate gradient checkpointing** such as in [A] and [B], so overall, we do not deem memory usage as a major bottleneck of our method.
>
> We will include this discussion into the revised paper as a limitation, as suggested.
>
> [A] Memory-Efficient Back-propagation Through Time (https://arxiv.org/abs/1606.03401)
>
> [B] Aligning Text-to-Image Diffusion Models with Reward Backpropagation (https://arxiv.org/abs/2310.03739v2)
>
> ### (W1) **Combination of existing ideas from ReSample, DPS and DCDP.**
>
> Our method is indeed motivated by the limitations of DPS and ReSample in terms of requiring a large number of sampling steps. However, we found that incorporating the proposed **network-regularized iterative backward consistency** not only improves quality but also significantly reduces the number of sampling steps and, consequently, the run-time.
>
> Furthermore, explicitly formulating the three consistency conditions may help clarify the individual effect of each condition and inspire future methods of a similar nature. We note that DCDP also does not utilize the network-regularized backward consistency.
>
> We refer the reviewer to Appendix C where we show the impact of each individual consistency on SITCOM’s performance. Further insights and relation to DPS and ReSample are given in Appendix D and Appendix E, respectively.
>
> ### (S3) **Arrows in the visualization of Figure 2 (left).**
>
> Thank you for pointing this out. We will fix it in the revised paper.
>
> ### (Q1) **Range of images before passing it to the measurement operator?**
>
> For natural images and MRI, the real and imaginary part, the range is [-1,1]. Thank you for pointing this out. We will include it in the revised paper.

---

> > ### Comment · Reviewer_Q3J7 · 2025-04-06
> >
> > I would like to thank the authors for their responses. I have read other reviews and responses to them. I think this is a good paper and I will maintain my score.

---

> > > ### Author Response · Authors · 2025-04-06
> > >
> > > We sincerely thank the reviewer for taking the time to read our responses to their comments and our replies to the other reviewers. We greatly appreciate their positive assessment.

---

### Official Review · Reviewer_VNrg · 2025-03-14

**Overall Recommendation:** 3

**Summary:**

This paper introduces a diffusion posterior sampling method, SITCOM, for solving inverse problems. SITCOM is designed to hold three consistency conditions during the diffusion sampling procedure to guide the generation. The backward consistency aims to ensure that the solution after data consistency optimization is a valid output of diffusion models. Experiments on several image inverse problems validate the effectiveness of SITCOM.

**Claims And Evidence:**

The authors claim three conditions to hold for diffusion-based inverse problem solvers and a new optimization-based sampler to enforce the three conditions, which are corroborated by the experimental results.

**Essential References Not Discussed:**

Some recent works on diffusion-based inverse problem solving are not discussed in this paper.

- Moufad et al. "Variational Diffusion Posterior Sampling with Midpoint Guidance." arXiv:2410.09945
- Janati et al. "Divide-and-Conquer Posterior Sampling for Denoising Diffusion priors." NeurIPS 2024.
- Wu et al. "Principled Probabilistic Imaging using Diffusion Models as Plug-and-Play Priors." NeurIPS 2024.
- Dou et al. "Diffusion Posterior Sampling for Linear Inverse Problem Solving: A Filtering Perspective." ICLR 2024.
- Cardoso et al. "Monte Carlo guided Denoising Diffusion models for Bayesian linear inverse problems." ICLR 2024.
- Wu et al. "Practical and Asymptotically Exact Conditional Sampling in Diffusion Models." NeurIPS 2023.

**Experimental Designs Or Analyses:**

The authors choose eight image inverse problems commonly studied in previous works. The experimental design is reasonable and demonstrates the effectiveness of the proposed algorithm.

**Methods And Evaluation Criteria:**

SITCOM optimizes the noisy sample $v_i$ at time $i\Delta t$ to satisfy the measurement constraints and perform a resampling step in each diffusion step. The proposed method is evaluated on inverse problems of natural images and MRI data, measured by common reconstruction metrics, PSNR, SSIM, and LPIPS.

**Other Comments Or Suggestions:**

- Page 6 looks too compact in its current form.

**Other Strengths And Weaknesses:**

#### Strengths

- This paper establishes three reasonable consistency conditions to guide the design of the algorithm.
- SITCOM demonstrates better performance in image restoration and MRI compared to existing methods.

#### Weaknesses

- The definition of backward consistency is confusing. It requires $\hat x_0^\prime$ to be the output of the diffusion model for some input $v_t$. It seems very hard to check whether this requirement is satisfied, given $v_t$ is an arbitrary input to the diffusion model. While SITCOM automatically satisfies this condition, it is not convincing that previous methods do not satisfy this condition just by looking at the definition.
- This paper is missing several recent references. It is possible that the comparison to these works will influence the position of this paper.

**Questions For Authors:**

- How are the hyperparameters $\delta, \gamma, \lambda, K$ determined? Does the performance of SITCOM depend on specific choices of these hyperparameters?
- Could the authors comment on the missing references, and why are these methods not included in the comparison?

- What are the reasons for SITCOM running faster than baseline methods given its additional overhead of resampling and multiple gradient updates?

Justification: I have some concerns about this paper as listed above, but I am open to revisiting my decision if the authors can address these concerns.

**Relation To Broader Scientific Literature:**

This paper may be of interest to a broader audience in other scientific domains that consider inverse problems with rich prior data.

**Theoretical Claims:**

The authors state three desirable consistency conditions, measurement, backward, and forward consistency, to solve inverse problems. The backward consistency is claimed to be new compared to previous literature. The backward consistency means a reconstruction $\hat x_0^\prime$ is the Tweedie output of the diffusion model for some input $v_t$. The proposed method indeed enforces three consistency conditions.

---

> ### Author Rebuttal · Authors · 2025-04-01
>
> We appreciate the reviewer's recognition that our paper establishes three reasonable consistency conditions to guide the algorithm's design and acknowledges SITCOM's superior performance in image restoration and MRI.
>
> We refer the reviewer to (https://anonymous.4open.science/r/SITCOM-B65F/rebuttal_table.md) for Tables C, D, and E.
>
> We thank the reviewer for their willingness to revisit their decision.
>
> ### (W1) **Backward consistency**
>
> Thank you for raising this question. The reviewer is essentially asking whether the estimate $\hat{x}_0$ produced by other methods (which do not explicitly enforce backward consistency) might satisfy backward consistency automatically, given that the space of possible $v_t$'s.
>
> Our response is that this scenario is unlikely. The key point is that the output space of $f(.;t,\epsilon_\theta)$ (Tweedie's network denoiser) at a fixed noise level $t$ is not the entire ambient space $\mathbb{R}^d$, but rather a small subset of it. Therefore, the probability that an arbitrary vector $\hat{x}_0$ produced by another method lies within this subset is low—unless the method is explicitly designed to ensure this consistency.
>
> This is not an intuitive claim but is supported by the following reasoning:
>
> 1. "Diffusion models encode the intrinsic dimension of data manifolds" ICML24. This paper demonstrates that for small noise levels, DMs approximate the normal bundle of the data manifold. Consequently, our Tweedie denoiser $f(.;t,\epsilon_\theta)$ at small $t$ approximates the tangent bundle. When the training data (such as imagenet) lies on a low-dimensional manifold, the output of $f(.;t,\epsilon_\theta)$ is therefore concentrated near this manifold—within a region whose thickness is proportional to the noise level $t$. This indicates that the denoiser’s output is constrained to a small subset of the full space.
>
> 2. "The intrinsic dimension of images and its impact on learning" ArXiv21. Table 1 of this paper estimates the intrinsic dimension of common datasets and confirms that it is significantly lower than the ambient dimension. For example, ImageNet's intrinsic dimension is reported to be approximately 43, which is much smaller than the raw pixel space dimension.
>
> Taken together, these results suggest that the trained DMs has an inherent regularizing effect: its outputs, especially at low noise levels, lie close to a low-dimensional manifold of the ambient space. Therefore, our backward consistency condition—requiring $\hat{x}0 = f(v_t;t,\epsilon_\theta)$ for some $v_t$ —acts as a meaningful constraint rather than a vacuous one.
>
> We refer the reviewer to the experiment in (https://anonymous.4open.science/r/SITCOM-B65F/experiment.png) to support our theoretical argument.
>
> ### (W2 & Q2) **Other baselines**
>
> We appreciate the reviewer for pointing out these references. In our evaluation, we compare SITCOM with 7 pixel-space and 3 latent-space baselines (all from 2023+), as well as 7 baselines for MRI. Baselines were chosen based on their strong performance on linear and non-linear inverse problems under measurement noise (Appendix K).
>
> Some references, such as MGPS (ICLR25), were very recent and not initially considered. To ensure SITCOM's position, we include comparisons in Table C for MGPS, FPS, and DCPS, noting that dashes indicate tasks not covered by these methods. These baselines do not report PSNR or SSIM, but SITCOM remains competitive.
>
> Table D compares SITCOM to PnP-DM, showing that SITCOM outperforms it in all tasks. PnP-DM did not report results on ImageNet.
>
> For MCG-Diff, we found no common settings (pre-trained model, noise, task, dataset), so due to time constraints, we could not compare it but will include results/discussion in the revised paper.
>
> TDS is a general conditional sampler applied to image inpainting on MNIST (not high-dimensional) and motif-scaffolding (not an imaging inverse problem). We will include this discussion in the revision.
>
> ### (Q1) **The determination of $\delta$, $\gamma$, $\lambda$, and $K$**
>
> We thank the reviewer for their comment. For $\delta$, it is computed based on the noise in the measurements. See the discussion in Appendix J.2.1. For the case of $K$ and $\lambda$, see Appendix J.1 and Appendix J.3, respectively.
>
> For the case of the learning rate $\gamma$, we use 0.01 (see Table 17 of Appendix K.1), which is the default for ADAM.
>
> To address the reviewer's concern, we conduct the ablation study in Table E. As observed, the results are very close around the default value. **We will include these results in the Appendix.**
>
> ### (Q3) **SITCOM running faster than baseline methods.**
>
> We agree that SITCOM requires iterative gradient updates to solve Eq.(S1), including backpropagation through the pre-trained DM. However, enforcing the three consistencies enables SITCOM to perform well with fewer sampling steps, leading to a shorter overall runtime than other baselines. For an intuitive explanation, see Remark D.2 in Appendix D.

---

> > ### Comment · Reviewer_VNrg · 2025-04-07
> >
> > Thanks for your detailed response. My major concern was mainly on the backward consistency proposed in Definition 3.1. I appreciate both the theoretical explanation and the experiment that illustrates the gap between a measurement-consistent image and a backward-consistent sample. It would also be interesting to see this experiment conducted on the intermediate sampling trajectories of previous methods.
> >
> > The baselines chosen seem reasonable according to the explanation, and the ablation studies in Appendix J are also satisfactory.
> >
> > Most of my earlier concerns have been addressed at this point, and I will therefore raise my score to 3 and recommend acceptance.

---

> > > ### Author Response · Authors · 2025-04-07
> > >
> > > We sincerely thank the reviewer for taking the time to read our response. We appreciate increasing the score, and recommending acceptance.
> > >
> > > We also thank the reviewer for suggesting that we repeat the backward consistency experiment on the intermediate sampling trajectories of previous methods. We will include this in the revised paper.

---

### Official Review · Reviewer_uV3o · 2025-03-14

**Overall Recommendation:** 3

**Summary:**

This paper systematically analyzes three necessary conditions—forward consistency, measurement consistency, and backward consistency—that enable accurate inverse problem solving using diffusion models. A sampler is proposed to enforce these conditions while reducing the required number of reverse diffusion steps. The method is extensively evaluated on five linear tasks, three nonlinear tasks, and one medical imaging task (in the appendix).

**Claims And Evidence:**

The paper makes two primary claims:

1. Forward consistency, measurement consistency, and backward consistency are three necessary conditions for accurately solving inverse problems with diffusion models.
2. By implicitly enforcing backward consistency through network regularization, the required number of reverse diffusion model steps is significantly reduced.

These claims are partially supported by empirical experiments.

**Essential References Not Discussed:**

I did not identify any critical missing references.

**Experimental Designs Or Analyses:**

The experiment design is sound and valid.

**Methods And Evaluation Criteria:**

The method is evaluated on five widely used linear inverse problems, three nonlinear tasks, and one medical imaging task. Standard evaluation metrics, including PSNR, SSIM, FID, and LPIPS, are used to assess performance.

**Other Comments Or Suggestions:**

None.

**Other Strengths And Weaknesses:**

Strength:

1. The paper is well-motivated and well-written, providing a systematic analysis of how different consistency conditions impact inverse problem solving. It also offers intuitive explanations for most design choices.
2. The proposed method achieves consistent improvements over most selected baselines across both linear and nonlinear tasks while requiring fewer inference steps.

Weakness:

1. My main concern is that the technical novelty of the paper appears limited. While the paper is the first to formally propose and explicitly formulate the three consistency conditions, each has already been individually explored in prior work. More specifically, forward consistency and measurement consistency have been widely used, while the main enhancement in this paper is the introduction of backward consistency through optimization in the input space of the Tweedie-network denoiser rather than the output space. Since the Tweedie-network denoiser serves as an implicit regularizer, an ablation study on this component should be included in the main paper.
2. The reported inference time for the proposed method is significantly lower than the baselines, but this comparison may be misleading because each baseline uses a different number of function evaluations (NFEs). A more informative way to present the results would be to plot performance (y-axis) against NFEs (x-axis) for each method as a curve.
3. The choice of baselines is confusing. For example, the paper does not report RED-Diff results on the nonlinear deblurring task. Based on my experience, RED-Diff should perform particularly well on this task. Could the authors provide insights into why this comparison was not included?
4. The paper doesn't provide theoriratical justification on three consistency condition in the proposed sampler.

**Questions For Authors:**

Please see above.

**Relation To Broader Scientific Literature:**

This work contributes to the field by systematically analyzing the role of three consistency conditions in determining reconstruction accuracy for inverse problems solved using diffusion models.

**Theoretical Claims:**

None.

---

> ### Author Rebuttal · Authors · 2025-04-01
>
> We thank the reviewer for their comments. We refer the reviewer to (https://anonymous.4open.science/r/SITCOM-B65F/rebuttal_table.md) for Tables A & B.
>
> We're glad the reviewer finds our paper well-motivated and well-written and acknowledges SITCOM's consistent improvements over most baselines with fewer inference steps. Our point-by-point response follows.
>
> ### (W1) **Ablation study on the implicit regularizatoin of Tweedie-network denoiser.**
>
> We appreciate the reviewer’s feedback. While forward and measurement consistencies have been explored before, our key contributions are: 1. **Backward consistency** is explicitly formulated and enforced through optimization. To our knowledge, this is the first work to introduce such a formulation and demonstrate its regularizing effect during sampling. 2. We integrate all three consistencies—forward, measurement, and backward—into a unified sampling framework, improving both reconstruction quality and efficiency. Notably, our approach reduces sampling steps without sacrificing accuracy, leading to computational savings.
>
> For the reviewer’s concern about ablation on the denoiser’s implicit regularization, see Appendix C.1, where we isolate the effect of backward consistency by comparing results with and without it. The findings show that enforcing backward consistency improves reconstruction accuracy and stability. As suggested by the reviewer, in the revised paper, we will include this part in the main body of the paper.
>
> Appendices C.2 and C.3 further provide ablations on measurement and forward consistency, respectively, supporting our claim that each component contributes meaningfully and coherently to the overall performance of SITCOM.
>
> ### (W2) **The inference time using number of function evaluations (NFEs).**
>
> We appreciate the reviewer’s question. All run-time results were obtained on the same RTX5000 GPU for fair comparison, with baselines run on our machine using their default settings, including NFEs.
>
> We agree that reporting NFEs is valuable and include this in Table A for super-resolution. Due to time constraints, we couldn't provide a full performance vs. NFE plot but offer partial results for insight.
>
> DPS and DDRM are competitive only at $N=1000$ (Figure 11, DPS paper). For DCDP, we found fewer NFEs suffice for linear tasks, while non-linear tasks still require many. Thus, we evaluated DCDP across a wide NFE range in Table 10 (Appendix I.2). SITCOM was similarly evaluated in Tables 10 (Appendix I.2) and 13 (Appendix J.1).
>
> Our results suggest reducing NFEs degrades baseline performance, though some methods are less sensitive. Additional experiments (Appendix I.2) show that under comparable runtime constraints, SITCOM achieves the best reconstruction quality (Tables 10, 11, and 12).
>
> ### (W3) **Choice of baselines and RED-Diff results on nonlinear deblurring.**
>
> We thank the reviewer for pointing this out. Table B shows comparison between RED-diff and SITCOM using ImageNet for non-linear deblurring (NDB) with measuement noise of 0.05.
>
> In our experiments, this method was not very good on LPIPS which we consider one of the most widely used perceptual metrics. As a result, we chose to compare against DAPS and DCDP for this task, as both achieved better LPIPS scores. The high PSNR reported in RED-diff's Table 1 (the ~45dB for NDB) is for the measurement noisless case.
>
> In general, the selection criteria for baselines is based on these baselines’ competitive performance on several linear and non-linear inverse problems under measurement noise (Appendix K).
>
> We would also like to point out that in Appendix I.2 (Table 11), we provide additoinal comparison results between SITCOM and RED-diff under time constraints using two linear tasks and one non-linear task.
>
> ### (W4) **Theoretical justification on the three consistency conditions.**
>
> We thank the reviewer for their comment. Enforcing forward consistency can pull intermediate solutions off the reversed diffusion trajectory (Figure 1), making establishing theoretical guarantees highly challenging. While we can show partial results, such as enforcing data consistency (up to a small error) using standard SGD convergence results, establishing overall error bounds for the final reconstruction remains difficult.
>
> Similar challenges exist in prior methods like ReSample, DCDP, DAPS, and DMPlug, which enforce data consistency at each step but lack overall reconstruction guarantees. The nonlinear nature of network regularization in our backward consistency further complicates the analysis.
>
> Nonetheless, Appendix D provides an intuitive explanation of why SITCOM approximates posterior samples similarly to DPS and why it may offer faster performance. We hope this analysis enhances understanding and confidence in our method.

---

### Decision · Program_Chairs · 2025-05-01

**Decision:**

Accept (poster)

**Comment:**

This paper handles three necessary conditions such as forward consistency, measurement consistency, and backward consistency for introducing better posterior sampling on solving inverse problems.

All three reviewers are convinced of papers' quality, especially on analysis of three conditions (uV3o, VNrg) and on clarity of supportive experiments (Q3J7). Even there are negative comments on the technical novelties (Q3J7,uV3o), computational burden (Q3J7), and missing references(VNrg), the authors resolved such concerns in rebuttal phase. Finally, three reviewers recommend accepts to this paper. I also agreed on reviewers' comments.

I recommend publication of this paper.